# An improved estimate of inorganic iodine emissions from the ocean using a coupled surface microlayer box model

Ryan J. Pound[1], Lucy V. Brown[1], Mat J. Evans[1,2], and Lucy J. Carpenter[1]

[1]Wolfson Atmospheric Chemistry Laboratories, Department of Chemistry, University of York, York, YO10 5DD, UK
[2]National Centre for Atmospheric Science, University of York, York, YO10 5DD, UK

**Correspondence:** Ryan J. Pound (ryan.pound@york.ac.uk)

**Abstract.** Iodine at the ocean's surface impacts climate and health by removing ozone ($O_3$) from the troposphere both directly, via ozone deposition to seawater, and indirectly via the formation of iodine gases which are released into the atmosphere. Here we present a new box model of the ocean surface microlayer that couples oceanic $O_3$ dry deposition to inorganic chemistry to predict inorganic iodine emissions. This model builds on the previous work of Carpenter et al. (2013), improving both chemical and physical processes. This new box model predicts iodide depletion in the top few micrometres of the ocean surface, due to rapid chemical loss to ozone competing with replenishment from underlying water. From this box model, we produce parameterised equations for HOI and $I_2$ emissions which are implemented into the global chemical transport model GEOS-Chem along with an updated sea surface iodide climatology. Compared to the previous model, inorganic iodine emissions from some tropical waters decrease by as much as half, while higher latitude emissions increase by a factor of $>>10$. With these large local changes, global total inorganic iodine emissions increased by $\sim 49\%$ (2.99 Tg to 4.48 Tg) compared to the previous parameterization. This results in a negligible change in average tropospheric OH ($<0.2\%$) and tropospheric methane lifetime ($<0.2\%$). The annual mean tropospheric $O_3$ burden decreases (-1.5% to 325 Tg), however, higher latitude surface $O_3$ concentrations decrease by as much as 20%.

*Copyright statement.* TEXT

## 1 Introduction

Iodine in the atmosphere and at the ocean-atmosphere interface is a large sink for tropospheric ozone ($O_3$). Dry deposition of $O_3$ to the ocean was thought to account for approximately one-third of the total $O_3$ loss to dry deposition (Ganzeveld et al., 2009), however, more recent work using more advanced representations of oceanic ozone dry deposition has revised this contribution down to $\sim 15\%$ (Luhar et al., 2018; Pound et al., 2020). At the ocean surface, the reaction between $O_3$ and iodide ($I^-$) is thought to represent a significant fraction of this loss (Fairall et al., 2007; Carpenter et al., 2013). Most global models have a simplistic representation of oceanic $O_3$ dry deposition, which contributes to the uncertainty in tropospheric $O_3$ (Ganzeveld et al., 2009; Hardacre et al., 2015). Including a more advanced oceanic dry deposition scheme which incorporates the chemical

loss of $O_3$ to $I^-$ along with the physical processes that control $O_3$ dry deposition has been shown to improve model comparisons to observations of both oceanic dry deposition velocity and remote marine surface $O_3$ concentrations (Luhar et al., 2017, 2018; Pound et al., 2020).

Photochemical cycling of iodine in the atmosphere leads to efficient chemical loss of $O_3$, perturbs $HO_x$ (Vogt et al., 1999; Alicke et al., 1999; Allan et al., 2000; Bloss et al., 2005) and along with other short-lived halogens emitted from the ocean surface have a substantial indirect impact on climate (Saiz-Lopez et al., 2023). Iodine compounds photolyze to produce atomic iodine (I) which is then rapidly oxidised by $O_3$ to form iodine oxide (IO). The dominant loss route is $IO + HO_2$ to return to HOI, which on photolysis leads to a net loss of $O_3$ (Sommariva et al., 2012; Saiz-Lopez et al., 2012). The inclusion of iodine emissions and subsequent chemistry into global chemistry transport models decreases tropospheric ozone concentration by 6-10% (Sherwen et al., 2016; Iglesias-Suarez et al., 2020; Pound et al., 2023b) with the largest impact being in the marine boundary layer (MBL) and coastal regions. IO can also impact both $HO_x$ ($OH + HO_2$) and $NO_x$ ($NO + NO_2$) concentrations (Sommariva et al., 2012; Sherwen et al., 2016). However, globally, iodine has a small impact on the atmospheric OH concentration. Whilst the reduction in $O_3$ by iodine reduces the primary chemical production of OH, iodine chemistry increases the conversion of $HO_2$ to OH, offsetting the reduction in primary production (Sherwen et al., 2016; Pound et al., 2023b).

Organic iodine species have been shown in laboratory experiments as the source of nucleation of new particles in coastal environments (Hoffmann et al., 2001). Recent work has also supported atmospheric iodine playing an important role in particle formation in the MBL, with the impact of iodine on aerosol formation and growth larger than previously thought (Huang et al., 2022). Combined with more efficient recycling of iodine from aerosol particles (Tham et al., 2021), this could mean that current global chemistry transport models underestimate the role of iodine in aerosol formation and its spatial range of impact.

Recent observations show that approximately 0.7 ppt of reactive iodine species are injected into the stratosphere, largely in the form of longer-lived organic iodine species and particulate iodine (Koenig et al., 2020). This has an important impact on stratospheric $O_3$, particularly in the tropical lower stratosphere (Saiz-Lopez et al., 2015). Based on these iodine levels reaching the stratosphere, recent model studies have shown that iodine can significantly impact the Antarctic $O_3$ hole, with iodine's role in modulating stratospheric $O_3$ likely to increase in relative importance as anthropogenic chlorine and bromine emissions decrease (Cuevas et al., 2022).

$I^-$ in the ocean is formed from the thermodynamically more stable iodate ($IO_3^-$) via biological reduction processes (Truesdale and Jones, 2000; Chance et al., 2007; Amachi, 2008; Wadley et al., 2020). $I^-$ and $IO_3^-$ combined represent the majority of the total iodine in the ocean. Due to the dependence on biological reduction, $I^-$ concentrations in the ocean could display sensitivity to both seasonal and climate timescales (Carpenter et al., 2021).

The sea surface microlayer (SML) covers the world's oceans to a significant extent, ranging in depth from 1-1000 $\mu$m and having distinct chemical and biological properties from underlying waters, and is the interface between the ocean and atmosphere (Wurl et al., 2011; Cunliffe et al., 2013; Wurl et al., 2017). Following the initial reaction between $O_3 + I^-$ in the top $\sim$ 3 $\mu$m of the SML (the reaction-diffusion length), further aqueous chemistry in the SML produces iodinated compounds which can subsequently be emitted into the atmosphere. The largest components of iodine emissions from the ocean surface are the inorganic compounds HOI and $I_2$, which are thought to contribute approximately 2 Tg yr$^{-1}$ of iodine to the global atmosphere (Carpenter et al., 2021). An additional 0.6 Tg yr$^{-1}$ of iodine arises from the emission of iodinated hydrocarbons ($CH_3I$, $CH_2I_2$, $CH_2IBr$ and $CH_2ICl$) (Jones et al., 2010; MacDonald et al., 2014; Prados-Roman et al., 2015).

The $O_3$ uptake rate by aqueous iodide solutions has been found to be significantly decreased by the addition of surfactants which form a monolayer across the solution and suppress exchange (Rouvière and Ammann, 2010). Laboratory studies of ozonised SML samples found that volatile organic iodine emissions were a negligible fraction of total iodine emissions (Tinel et al., 2020). The addition of organic material has also been found to suppress $I_2$ emissions from $I^-$ solutions, with this largely being attributed to a decrease in the net transfer of $I_2$ from the aqueous to gas phase (Reeser and Donaldson, 2011; Shaw and Carpenter, 2013; Tinel et al., 2020). Modelling studies of IO in the Indian Ocean needed to reduce inorganic iodine emissions by 40% to reasonably match cruise-based observations from the region (Mahajan et al., 2021).

Anthropogenic activity has contributed to increased iodine emissions since preindustrial times (Cuevas et al., 2018; Legrand et al., 2018; Saiz-Lopez et al., 2023), largely due to increased tropospheric $O_3$ increasing inorganic iodine emissions from the ocean. The increase in anthropogenic emissions from preindustrial to the present day has also shifted the partitioning of inorganic halogens from reactive to reservoir species (Barrera et al., 2023). Model studies using future climate scenarios forecast a key role of iodine in $O_3$ destruction through the 21st century (Badia et al., 2021).

Carpenter et al. (2013) created a kinetic box model of the SML to predict inorganic iodine emissions which were parameterised as functions of surface $O_3$ concentration, $I^-$ concentration in the ocean and wind speed. This model predicts exponentially increasing inorganic iodine emissions as wind speed decreases due to an increasing fraction of iodine being emitted to the atmosphere as opposed to being mixed with the underlying water. As such, a minimum wind speed of 5.5 ms$^{-1}$ was applied when implemented in the global chemistry transport model GEOS-Chem (Sherwen et al., 2016). This SML model also does not directly couple the SML chemistry to $O_3$ dry deposition and as such is unable to capture feedback between $O_3$ deposition, $I^-$ depletion in the SML (Schneider et al., 2020), and the chemical production and emission of inorganic iodine compounds. Finally, the equations provided by Carpenter et al. (2013) did not include a temperature dependence. In reality, there are a host of temperature-dependent processes involved in iodine emissions including the $O_3 + I^-$ reaction (Brown et al., 2024), the diffusivity of $O_3$ (Johnson and Davis, 1996) and the solubility of HOI and $I_2$.

Several experiments have measured the rate constant of the $O_3 + I^-$ reaction at a single temperature (Garland et al., 1980; Hu et al., 1995; Liu et al., 2001). A temperature-dependent rate by Magi et al. (1997) has been used in previous work to model oceanic $O_3$ dry deposition (Luhar et al., 2017, 2018; Pound et al., 2020) and inorganic iodine emissions (Carpenter et al., 2013). However, the Magi et al. (1997) laboratory study used iodide concentrations of 0.5-3.0 M, which are substantially higher than the typical ocean surface range of 10-100 of nM (Chance et al., 2014). At $I^-$ concentrations above 1000 nM, the reaction between $O_3$ and $I^-$ occurs at the water surface (Moreno et al., 2018). However, with environmental concentrations of $I^-$, 10-100 nM, the reaction mainly occurs by a different mechanism, within the bulk aqueous phase (Moreno et al., 2018). Ozone uptake experiments under environmentally comparable $I^-$ concentrations also support the aqueous reaction dominating the $O_3 + I^-$ reaction (Schneider et al., 2020; Brown et al., 2024). Recently, Brown et al. (2024) have calculated a new temperature-dependent rate for $O_3 + I^-$ under environmentally comparable iodide concentrations (100-10000 nM) and an $O_3$ mixing ratio of 40 ppb at 1 atm. A range of temperatures from 288-303 K were applied, yielding a temperature dependence which can be applied to the interaction of ozone and iodide in the ocean surface. This rate is comparable to other experimental results which did not sample a range of temperatures (Garland et al., 1980; Liu et al., 2001; Hu et al., 1995).

Here we propose a new air-SML-ocean exchange model to couple the processes of oceanic ozone dry deposition to inorganic iodine emissions, which incorporates recent advancements in inorganic iodine chemistry. Section 2 describes the construction of the model and the equations used to describe the physical and chemical components. Sections 3 to 5 diagnose the model's sensitivity to mixing, rate of $O_3 + I^-$ reaction, and salinity and inter-halogen reactions. We then compare this new model to the existing model from Carpenter et al. (2013) (section 7) and experimental results (section 8). Finally, parameterised functions for estimating HOI and $I_2$ emissions calculated by this coupled ocean-atmosphere exchange model (section 9) are implemented in a global chemical transport model (GEOS-Chem Classic) to give a new estimate for global inorganic iodine emissions and their impact on tropospheric $O_3$ (section 10).

## 2 SML box model description

Figure 1 shows a simplified overview of the ocean-atmosphere exchange model described in this paper. Ozone deposition is based on the resistance-in-series scheme and is further described in section 2.1. The SML is composed of many sublayers, defined by their physical, chemical or biological properties (Soloviev and Lukas, 2014; Carpenter and Nightingale, 2015). This model considers only the top several micrometres of the SML, defined as the reaction-diffusion layer where the rate of chemical loss of $O_3$ dominates over turbulent transfer (Luhar et al., 2018). This length scale is dependent on the molecular diffusivity and chemical reactivity of $O_3$ and given by equation 8. The chemistry scheme employed is described in section 2.2. Finally, the mixing of the SML with the atmosphere and the ocean is based on the method described by Cen-Lin and Tzung-May (2013) which is described in sections 2.3.1 and 2.3.2 respectively. Table 1 lists all the inputs, outputs and variables used to calculate the flux of ozone into the SML and the emission of inorganic halogens from the SML.

The model was developed in Python using Cantera as the chemistry solver (Goodwin et al., 2022). The model presented here also uses functions from SciPy (Virtanen et al., 2020), Pandas (development team, 2020), and NumPy (Harris et al., 2020). Calculations of the salinity and temperature-dependent unitless Henry's law ($H$), Schmidt number in the air ($S_{ca}$) and water ($S_{cw}$), airside ($k_a$) and waterside ($k_w$) transfer velocities are calculated using the recommended functions from Johnson (2010). The model runs presented here used a physical timestep of $1 \times 10^{-4}$ s and typically reaches equilibrium within 4 s.

## 2.1 Coupled ozone dry deposition

Ozone dry deposition velocity ($v_d$) is calculated using the resistance-in-series scheme based on Wesely and Hicks (1977), equation 1, this is then used to calculate the flux of ozone into the ocean surface microlayer. Airside resistances that represent turbulent transport to the surface ($r_a$) and transport through the atmospheric quasilaminar sub-layer, which is the air directly above the surface microlayer ($r_b$), are calculated using equations 2 and 3 respectively (Chang et al., 2004).

$$v_d = \frac{1}{r_a + r_b + r_c} \tag{1}$$

$$r_a = \frac{u_{10}}{u^{*2}} \tag{2}$$

$$r_b = \frac{5}{u^*} S_{ca}^{2/3} \tag{3}$$

Where $u$ is the 10m wind speed with units of $\mathrm{ms}^{-1}$, $u^*$ is the friction velocity with units of $\mathrm{ms}^{-1}$, and $S_{ca}$ is the Schmidt number of $O_3$ in air and is calculated using the method from Tsilingiris (2008) as recommended by Johnson (2010).

The surface resistance ($r_c$) captures the chemical and physical processes in the SML which control ozone loss. We employ the two-layer method of Luhar et al. (2018) to calculate $r_c$ which is shown in equation 4.

$$r_c = \frac{1}{\alpha\sqrt{aD}} \left[ \frac{\Psi K_1(\xi_\delta)\sinh(\lambda) + K_0(\xi_\delta)\cosh(\lambda)}{\Psi K_1(\xi_\delta)\cosh(\lambda) + K_0(\xi_\delta)\sinh(\lambda)} \right] \tag{4}$$

The terms $\xi_\delta$, $\Psi$, $\lambda$ are given in equations 5, 6 and 7 respectively.

$$\xi_\delta = \left[ \frac{4a}{\kappa u_w^*} \left( \delta_m + \frac{D}{\kappa u_w^*} \right) \right]^{\frac{1}{2}} \tag{5}$$

$$\Psi = \left[ 1 + \left( \frac{\kappa u_w^* \delta_m}{D} \right) \right]^{\frac{1}{2}} \tag{6}$$

$$\lambda = \delta_m \sqrt{\frac{a}{D}} \tag{7}$$

where $u_w^*$ (ms$^{-1}$) is the water-side friction velocity, $\delta_m$ is the thickness of the reaction-diffusion layer of the sea-surface microlayer (m) calculated using equation 8 (Luhar et al., 2017). $K_0$ and $K_1$ are modified Bessel functions of the second kind with order zero and one respectively, and $\kappa$ is the von Kármán constant ($\approx 0.4$). $a$ is the chemical reactivity of $O_3$ with $I^-$ (defined in equation 9). $a$ uses the second order rate-coefficient ($k$) from either Magi et al. (1997) or Brown et al. (2024) with units of M$^{-1}$s$^{-1}$. The diffusivity $O_3$ in the SML ($D$, m$^2$s$^{-1}$) is from Johnson and Davis (1996) and shown in equation 10. This calculation of $D$ does not account for the impact of organics (particularly surfactants) which will impact the transfer of $O_3$ into the SML, this model is currently limited to inorganic chemistry and the limitations of this are discussed further in 11. $\alpha$ is the dimensionless solubility of $O_3$ from Morris (1988) shown in equation 11

$$\delta_m = \sqrt{\frac{D}{a}} \tag{8}$$

$$a = k[I^-] \tag{9}$$

$$D = 1.1 \times 10^{-6} exp\left(\frac{-1896}{T}\right) \tag{10}$$

$$\alpha = 10^{-0.25-0.013(T-273.16)} \tag{11}$$

The dry deposition velocity ($v_d$) is coupled to the SML chemistry via the $I^-$ concentration and is recalculated as the model advances in time towards equilibrium.

## 2.2 Chemistry

The aqueous inorganic halogen chemistry scheme used in this model is shown in Table 2. Here we employ an extended version of the iodine chemistry scheme used by Carpenter et al. (2013) and similar to that of Schneider et al. (2023) with the addition of further inter-halogen reactions involving bromine and chlorine species. A further difference between this chemistry scheme and that of Carpenter et al. (2021) is that we explicitly include the reaction step of $O_3$ + $I^-$ producing $IO^-$ (R1a) rather than HOI directly, alongside its subsequent conversion to HOI (R7). This has little impact on the total simulated inorganic iodine emissions as $IO^-$ quickly reacts to form HOI at ocean pH, but presents a more complete representation of the chemistry.

To explore the sensitivity of total iodine emissions to the rate coefficient of the $O_3 + I^-$ reaction, two different forms of the temperature-dependent rate coefficient are used. The first of these (reaction R1a from table 2) uses the rate published by Magi et al. (1997) which is also the rate used by Carpenter et al. (2013) in their model. The second rate (reaction R1b from table 2) is the more recent temperature-dependent rate from Brown et al. (2024) which has a much weaker temperature dependence than that of Magi et al. (1997). The different temperature dependencies of these two rates are shown in figure 2.

## 2.3 Mixing processes

### 2.3.1 Emissions of inorganic iodine

The net flux of a species from the SML into the atmosphere ($F_a$) is calculated from the concentration in the SML ($C_{sml}$) and the concentration in the atmosphere ($C_a$, equation 12). Atmospheric fluxes are calculated for HOI, $I_2$, IBr and ICl, HOCl, HOBr, $Br_2$, $Cl_2$, BrCl. HOI and $I_2$ have the largest fluxes, with the other species emitted in negligible amounts due to their high solubility and relatively low concentrations in the SML.

$$F_a = k_a(H * C_{sml} - C_a) \tag{12}$$

$k_a$ is calculated following the recommendation from Johnson (2010) using equation 13

$$k_a = + \frac{u_*}{13.3 S_{ca}^{0.5} + C_D^{-0.5} - 5 + \frac{ln(S_{ca})}{2\kappa}} \tag{13}$$

where $u_*$ is the friction velocity ($ms^{-1}$), $C_D$ is the drag coefficient ($ms^{-1}$) from Smith (1980)

$$10^3 C_D = 0.61 + 0.063 u_{10} \tag{14}$$

### 2.3.2 Ocean mixing with SML

This model employs SML concentrations mixing with the bulk ocean concentration ($C_b$) on two timescales and follows the approach used by Cen-Lin and Tzung-May (2013). The first mixing process, molecular transfer, occurs on the order of 0.1 - 1 seconds and is given by equation 15.

$$F_b = Rk_w(C_b - C_{sml}) \tag{15}$$

where R accounts for the effects of surfactants suppressing the transport between the SML and bulk ocean. A value of 0.9 is used in this study to represent the open ocean (Goldman et al., 1988; Frew et al., 1990; Cen-Lin and Tzung-May, 2013). $k_w$ is calculated using equation 16 which follows the recommendations of Johnson (2010) in using the Nightingale et al. (2000) approach. $u_{10}$ is the 10m wind speed, $S_{cw}$ is the Schmidt number of the gas in water and $S_{c600}$ Schmidt number of $CO_2$ at 20 °C.

$$k_w = (0.222u_{10}^2 + 0.333u_{10})(\frac{S_{cw}}{S_{c600}})^{0.5} \tag{16}$$

The second mixing process is surface renewal, representative of larger scale eddy mixing, and is given by equation 17. It is a significantly slower process than the mixing described above and is typically of the order of minutes or longer, but has been included for completeness. Surface renewal and the suppression of transfer velocity by surfactants ($R$) are new developments in this model compared to Carpenter et al. (2013).

$$F_r = (3.42 \times 10^{-3}u_{10} + 2.7 \times 10^{-3})(C_b - C_{sml}) \tag{17}$$

Fluxes for mixing between the SML and bulk ocean are calculated for HOI, $I_2$, $O_3$, IBr, ICl, $IO_3^-$, HOBr, HOCl, $Br_2$, $Cl_2$, BrCl, $I^-$, $Br^-$ and $Cl^-$. The mixing processes described here are only representative of passive diffusion and do not take into account any electrostatic effects. Solutions with ions have been found to have stronger electric fields at the air-water interface than within the bulk due to charge separation and this can contribute to an increased concentration of ions and enhanced reaction rates (Xiong et al., 2020; Hao et al., 2022). However, given that the fast turbulent-driven mixing of the SML with the bulk water and the chemical depletion of $I^-$ within the SML occur on timescales of seconds or less under typical conditions, we consider additional effects which could impact the enhancement or depletion of $I^-$ within the SML are likely secondary. The control of $I^-$ in the SML by the equilibrium of chemical and physical processes represents a significant difference between this work and that of Carpenter et al. (2013), where it was prescribed as a constant. The impact of this difference is explored in section 3.

The concentration of $I^-$ is set based on the conditions being studied by the model, unless otherwise stated the sensitivity studies presented here use a concentration of 100 nM. $Br^-$ concentrations are set at 0.86 mM and $Cl^-$ at 0.55 M to replicate typical ocean salinity. The oceanic concentration of $IO_3^-$ is set at 200 nM (Chance et al., 2020). All other species are assumed to have zero bulk oceanic concentrations. $H^+$ and $OH^-$ are not subject to mixing and are manually set at each time-step to maintain a constant pH of 8.

## 3  Depletion of SML iodide

One difference between this and previous work is the model prediction of depletion of $I^-$ in the SML at low wind speeds (figures 3 and 4) due to its reaction with $O_3$. This is a direct consequence of the slow replenishment of $I^-$ in the SML from mixing with bulk water rather than being a fixed quantity as in previous models. Depletion of $I^-$ has been previously detected experimentally in artificial seawater (Schneider et al., 2023). The effect of this depletion on total inorganic iodine emission and the composition of that emission is shown in figure 3. To a lesser extent, depletion of $I^-$ is also greater at higher SST, as shown

in figure 4; this is entirely driven by the temperature dependence of the $O_3 + I^-$ reaction.

The depletion of $I^-$ accounts for roughly an 11% reduction in total inorganic iodine emissions at $2\ ms^{-1}$ wind speed, 285 K, 30 ppb $O_3$, and 100 nM of $I^-$ in bulk water (figure 3). The reduction in SML $I^-$ concentration also reduces $O_3$ dry deposition velocity by 15% under the same conditions.

## 4  Sensitivity to the temperature dependence of the $I^-$ + $O_3$ reaction

Here we compare two temperature-dependent rate constants for the $I^-$ + $O_3$ reaction. The first of these is that of Magi et al. (1997), which has been used in the previous model of Carpenter et al. (2013). We compare this to the more recent rate from Brown et al. (2024) which was derived from experiments conducted at $I^-$ concentrations of $\sim$100 - 10000 nM. The difference in the temperature dependence of total inorganic iodine emission is shown in figure 5. The newer rate constant results in substantially higher total inorganic iodine emissions at low SST. At 285K, total inorganic iodine emissions increase by $\sim$130% when using the rate coefficient from Brown et al. (2024) compared to Magi et al. (1997) (Figure 5a), and $O_3$ dry deposition velocity increases by 36%. Both increases are offset by $I^-$ enrichment decreasing from $\sim$96% to $\sim$90% at the same temperature (Figure 5c). Increased depletion of $I^-$ in the SML also results in the production pathways of $I_2$ from HOI becoming less competitive, resulting in the amount of $I_2$ emission relative to HOI decreasing (Figure 5b). At higher temperatures (above $\sim$25°C, figure 2), the Brown et al. (2024) rate is slower than that of Magi et al. (1997), resulting in decreased HOI production, however this is somewhat offset by the sensitivity of the model to the reaction-diffusion length which is dependent on this rate and is explored in section 6. All subsequent experiments using the box model use the Brown et al. (2024) rate due to it better reflecting the $O_3+I^-$ reaction under oceanic conditions.

## 5  $I_2$ emission salinity sensitivity

The experimental work of MacDonald et al. (2014) found a strong increase in $I_2$ emission at higher salinity, which was replicated in their accompanying model results. We also predict a positive salinity dependence on $I_2$ emissions in our base model (figure 6). The increase in $I_2$ emissions is partly from the additional chemical pathway to produce $I_2$ via ICl as concluded by MacDonald et al. (2014) (reactions R13, R15 and R17 in table 2). Additionally, the changes in solubility (due to salinity due to the salinity dependence of $H$ and $S_{cw}$, salting out effect), increase the total iodine emissions, shown by the difference in green and yellow lines in figure 6 where the chloride concentration was set to achieve a salinity of 35 PSU but the chlorine chemistry was removed from the chemical mechanism. The increase in total iodine emissions from the increase in $I_2$ emission has a negligible impact on HOI emissions, as HOI is in excess in the SML (Carpenter et al., 2013).

The largest increase in $I_2$ emission with the addition of salinity is observed in low turbulence conditions (low wind speed, figure 6b); here the effects on solubility have a larger effect than the additional chemical pathway to $I_2$ production provided by $Cl^-$. Under the conditions used in this study, $I_2$ emissions are increased by $\sim$150% with the addition of $Cl^-$. However, this is less than the 250% increase observed by MacDonald et al. (2014). Differences between experimental results and this model are discussed further in section 8. Figure 6 shows that similar to chloride, increasing bromide increases total $I_2$ production. This increase is the result of the additional pathway via IBr to produce $I_2$ (reactions R12 and R14 in table 2).

In contrast to these results, more recent work from Tinel et al. (2020) and Schneider et al. (2023) did not find an increase in $I_2$ emissions from increasing $Cl^-$, instead, $I_2$ emissions were suppressed in saline samples compared to just iodide solutions. Schneider et al. (2023) found that their results could be replicated by shifting the equilibrium for reaction R15a in table 2 to favour $I_2Cl^-$ over $I_2$ (reaction R15b in table 2). The result of implementing that change to the equilibrium in this model is shown in figure 7; the rate of $I_2$ production through the additional chemical pathway provided by ICl is reduced and iodine emissions decrease by up to 5%. Depletion of $I^-$ is unaffected.

## 6 Inorganic Iodine emission sensitivity to reaction-diffusion layer depth

Here we explore the sensitivity of predicted iodine emission fluxes to the depth of the reaction-diffusion layer ($\delta_m$). The values of $\delta_m$ calculated are typically between 2.2-2.9 $\mu$m for the conditions shown in figures 8 and 9 (using equation 8 and the Brown et al. (2024) rate constant). $\delta_m$ is directly dependent on SML temperature via $O_3$ diffusivity $D$ and the chemical reactivity $a$, with $a$ also giving $\delta_m$ a direct dependence on $[I^-]$. There is an indirect dependence between $\delta_m$ and wind speed due to the depletion of iodide in the SML which increases in low turbulence conditions. Under conditions with higher turbulence ($>3$ ms$^{-1}$), larger $\delta_m$ values increase total inorganic iodine emissions. However, under less well-mixed conditions, the coupling of mixing and the availability of $I^-$ in the SML creates a more complex relationship between total inorganic iodine emission and $\delta_m$. Further improvements to model predictions of total inorganic iodine emissions are therefore dependent on our understanding of the reaction-diffusion length and uncertainties in both $D$ and uncertainties in the rate of $O_3 + I^-$ (reaction R1).

## 7 Comparisons to existing model

Figure 10 compares the total iodine emissions predicted in this work to that of Carpenter et al. (2013) across wind speed, iodide, ozone and temperature ranges. The new model uses the $O_3 + I^-$ rate from Brown et al. (2024) and the updated equilibrium of reaction R15b from Schneider et al. (2023). Temperature dependence was not included in the Carpenter et al. (2013) equations. Two versions of the Carpenter et al. (2013) model are used in the wind speed comparison. The first is the equations as presented (solid black line), the second has a minimum wind speed of 5.5 ms$^{-1}$ applied (dashed black line, as used in the global modelling study of Sherwen et al. (2016)). For total iodine emission at the highest wind speeds, the new model tends

towards the old model. This is due to the efficient mixing at these higher wind speeds resulting in negligible $I^-$ depletion in the SML, thus more closely resembling the old model which included a constant $I^-$ concentration in the SML. As wind speed

decreases, the two models diverge in their prediction of total iodine emission with the new model predicting less emission flux than the capped and uncapped (Carpenter et al., 2013) model. This decrease is most notable at very low wind speeds where the new model tends towards no iodine emission as wind speed tends towards zero, rather than the Carpenter et al. (2013) model where total iodine emissions exponentially increase as wind speed tends to zero.

Carpenter et al. (2013) found a simple multiplicative and approximately linear relationship between $O_3$ concentration and total iodine emission. The slightly dampened relationship between $O_3$ and total iodine emission in this model is likely because higher $O_3$ concentration causes a greater depleting effect on SML $I^-$ concentration, reducing total iodine emission. The new model predicts a similar trend of $I^-$ dependence of total iodine emission to Carpenter et al. (2013). Additionally, the new model predicts that $I_2$ contributes a larger percentage of total iodine emissions, despite the change made to the chemistry scheme to

reflect lower $I_2$ emissions under oceanic salinity. This difference is likely due to a reduced HOI emission flux from the SML, resulting in more of the aqueous HOI remaining in the SML which can subsequently produce more $I_2$.

## 8    Comparisons to experimental data

Table 3 compares published experimental results for $I_2$ fluxes to predictions from this model under conditions that match those

used in the various experimental studies. One limitation in this comparison is replicating the waterside turbulence due to stirring (or not) of the aqueous solution and from the gas flow over the solution. Experimental setups which do not stir the solution have very different dynamics to the ocean surface, which this model has been designed to replicate. Additionally, some experimental results use $O_3$ and $I^-$ concentrations significantly higher than typical environmental conditions due to measurement instrument sensitivity (table 3). The model significantly underestimates experimental results where the solutions were not stirred, possi-

bly due to a high depletion of surface iodide under such conditions which reduces the potential for gaseous iodine emissions (Schneider et al., 2023). For stirred experiments, however, the model predicts a similar range of $I_2$ emissions as the experiments.

## 9    Parameterised equation for HOI and $I_2$ emission flux

Here we present two mathematical functions to predict HOI (equation 18) and $I_2$ (equation 19) emission fluxes based on [$O_3$],

bulk [$I^-$], wind speed and SST. A non-linear least squares fit was used on 5000 unique combinations of model inputs covering environmentally comparable ranges of each variable (5-60 ppb of $O_3$, 0.1-11.1 ms$^{-1}$ wind speed, 20-240 nM bulk [$I^-$] and 274-300 K SST). All other parameters are kept constant in the sensitivity analysis. The model sensitivity studies are run using the $O_3 + I^-$ rate from Brown et al. (2024) and the updated equilibrium of reaction R15b from Schneider et al. (2023). These

equations have a high correlation with the results from the SML box model, $R^2 = 0.92$ for HOI and $R^2 = 0.92$ for $I_2$, and no strong bias in over or underestimating the model results (figure 11).

$$HOI = 6.9 \times 10^{-11} \left( \frac{u + 6.2}{12} \right) e^{-0.034T - \left( \frac{u+6.2}{12} \right)^2} [O_3]_g^{0.92} [I^-]^{0.64} \tag{18}$$

$$I_2 = 4.2 \times 10^{-19} \left( \frac{u + 3.1}{7.2} \right) e^{0.011T - \left( \frac{u+3.1}{7.2} \right)^2} [O_3]_g^{0.73} [I^-]^{1.5} \tag{19}$$

where HOI and $I_2$ emission are in kg m$^{-2}$ s$^{-1}$, $T$ is the sea surface temperature (K), $u$ is the 10 m wind speed $(ms^{-1})$, $[I^-]$ is the bulk water iodide concentration (nM), and $[O_3]_g$ is the atmospheric ozone mixing ratio (ppb).

The most notable difference between the parameterised equations presented here and those from Carpenter et al. (2013) is the inclusion of $T$ as a parameter. Carpenter et al. (2013) found a linear relationship between both HOI and $I_2$ emissions and atmospheric $O_3$. This relationship with $O_3$ is reduced for HOI which can be attributed to the impact of $I^-$ depletion in the SML being increased at higher $O_3$ concentrations, reducing the rate at which the $O_3 + I^-$ reaction can occur. The effect of $O_3$ concentration on $I^-$ depletion is further enhanced for $I_2$ production (hence a further reduction in the $I_2$ dependence on $O_3$ concentration) as the subsequent chemical pathways to convert HOI to $I_2$ also depend on the availability of $I^-$ in the SML. $I^-$ depletion can also explain the increase in dependence of HOI emission on SML $[I^-]$ (from a power of 0.5 to 0.63) and $I_2$ (from a power of 1.3 to 1.5) due to a relative increase in the availability of $I^-$ in the SML at higher bulk water concentrations.

## 10    Implementing the new iodine emission equations in GEOS-Chem Classic

We use the GEOS-Chem classic model (Bey et al., 2001) version 14.1.1 (GCC14.1.1, 2023) for global modelling of inorganic iodine emissions and their impact on tropospheric composition. GEOS-Chem Classic is a chemical transport model with a HOx-NOx-VOC-$O_3$-halogen-aerosol tropospheric chemistry scheme. The current version of the halogen chemistry scheme is described by Wang et al. (2021), with organic iodine emissions based on Ordóñez et al. (2012), and inorganic iodine emissions based on Carpenter et al. (2013) (as implemented by Sherwen et al. (2016)). The current inorganic iodine emissions in GEOS-Chem use surface oceanic iodide concentrations based on MacDonald et al. (2014), which under-predicts compared to observations (Sherwen et al., 2019) and differs from the iodide field used in calculating $O_3$ dry deposition which uses Sherwen et al. (2019) (Pound et al., 2020). Here, we implement the equations for inorganic iodine emission presented in section 8 and compare the impact of changing the iodide used from MacDonald et al. (2014) to the up-to-date, machine learning-derived, iodide climatology from Sherwen et al. (2019) for both inorganic iodine emissions providing symmetry with the current use of this iodide climatology in $O_3$ dry deposition.

A global spatial resolution of $4°$x$5°$ on the standard vertical grid (72 vertical levels) is used, running with chemistry in both the troposphere and stratosphere. Meteorological data is from MERRA-2 (Gelaro et al., 2017). Three model runs were con-

ducted which were identical in configuration, apart from inorganic iodine emissions. The first uses the default inorganic iodine emissions based on the equations from Carpenter et al. (2013) driven by MacDonald et al. (2014) oceanic $I^-$. The second uses the new iodine emission equations presented in this work (equations 18 and 19) driven by MacDonald et al. (2014) oceanic $I^-$. The third uses the new iodine emission equations driven by Sherwen et al. (2019) oceanic $I^-$. All other emission and time-step configurations were left at their recommended settings. Model simulations were conducted from the $1^{st}$ of July 2019 to the $1^{st}$ of July 2021, with the first year of the simulation considered a "spin-up" to allow the model to reach equilibrium. Although the 2020-2021 period is within a La Niña event, a change in temperature of 1K changes total inorganic iodine emissions by $\sim 3\%$ (figure 10). As such temperature variations due to ENSO are likely to result in changes in the inorganic iodine concentrations of less than 10% locally, and likely less globally.

The new iodine emission equations decrease the total global inorganic iodine emissions from 2.84 Tg yr$^{-1}$ to 2.78 Tg yr$^{-1}$, a decrease of 2%. Additionally, there is a slight increase in the ratio of $I_2$ emissions compared to HOI, with $I_2$ now accounting for 5.9% (previously 5.7%), however, this difference would not change the impact of iodine on the troposphere as both HOI and $I_2$ rapidly photolyse. Whilst this is a relatively small global total change, there is a significant re-distribution of total iodine emissions, with emissions from equatorial waters decreasing and high latitude emissions increasing, as shown in figure 12a. The changes in global emission distribution are largely driven by the change from Magi et al. (1997) to Brown et al. (2024) rate constant, with large increases in high latitude waters also being the result of the base model predicting near zero emissions in cold waters with low $[I^-]$. Combining the new iodine emission equations with the Sherwen et al. (2019) (figure 12b) iodide climatology results in an additional factor of $\sim$4 increase in total inorganic iodine emission from high latitude waters, decreasing to $\sim$1.5 outside of these regions. This iodide climatology increases the total global inorganic iodine emissions to 4.5 Tg yr$^{-1}$ (+49%). Due to the substantially improved comparison with observations, the iodide climatology from Sherwen et al. (2019) will be used in the following analysis.

The change in the distribution of inorganic iodine emissions substantially increases high latitude IO (and IO$_x$) concentrations, with percentage changes of >1000% as the base case model predicts very low or no IO concentration in these regions (figure 13). Equatorial IO has small regional increases and some decreases, mirroring the negligible increases and localised decreases in inorganic iodine emissions from warm waters. However, despite large regional changes, the change in area-weighted mean vertical iodine speciation is minimal, with large percentage changes reflecting small absolute increases. Figure 14 compares published observations of average daytime surface IO mixing ratios to the model predictions from the corresponding day of the year during the simulated period. This comparison only considers open ocean observations, coastal observations of IO have large influences from macro-algae emissions (Saiz-Lopez and Plane, 2004) which are not included in this model. The change in iodine emissions has little effect on the average model root mean square error of atmospheric IO, increasing it from 0.48 to 0.62 ppt (0.43 ppt to 0.58 excluding polar observations) with a change in the relative mean bias from -0.43 ppt to 0.43 ppt (-0.07 ppt to 0.55 ppt excluding polar observations), suggesting that there are still uncertainties in other aspects of our understanding of the iodine system such as the formation of HIO$_3$ (He et al., 2021; Huang et al., 2022) and the

photolysis scheme currently used for higher iodine oxides (Sherwen et al., 2016). Work to further understand the atmospheric

chemistry of iodine is still required if we are to have confidence in the predictions of our models. Wang et al. (2021) found less disagreement between their model and observation comparisons, however, that study included sea salt debromination which has a large impact on tropospheric $O_3$ and is by default deactivated in version 14.1.1 of GEOS-Chem. The increase in high latitude oceanic emissions of HOI and $I_2$ reduces the model error at the two Antarctic locations (Bharati and Maitri bases) and the MOSAiC Arctic observations included in figure 14, however, the model still significantly underestimates IO levels in the

Antarctic region. Atmospheric iodine observations made in the Antarctic region have been shown to have a source from sea ice (Saiz-Lopez et al., 2007; Atkinson et al., 2012); and a direct source of atmospheric IO from the snowpack (Frieß et al., 2010), these processes are not currently represented in the model.

Despite the large increase in total inorganic iodine emissions, there is only a 1.5% decrease in the tropospheric $O_3$ burden

(from 330 Tg to 325 Tg). As with IO, there are larger regional changes in both surface and zonal $O_3$, as shown in figure 15. Tropospheric $O_3$ at higher latitudes is decreased with the largest absolute and percentage changes above the Southern Ocean, while $O_3$ above the equatorial Indian Ocean and western mid-Pacific increases.

Figure 16 shows the impact of the change in iodine emissions on surface ozone predictions. For this, we compare the model

to a selection of surface ozone measurements from several World Meteorological Organization (WMO) Global Atmosphere Watch sites around the world (GAW; http://www.wmo.int/pages/prog/arep/gaw/gaw_home_en.html, accessed through EBAS http://ebas.nilu.no/, the database infrastructure operated by NILU – Norwegian Institute for Air Research).

At the northern high latitudes, we compare to $O_3$ observations made in Greenland (panel a of figure 16). The model dis-

410 agreement at this site, measured using root mean square error (RMSE), increases from 3.7 ppbv to 3.8 ppbv (2.7%), however, the model remains unable to replicate springtime $O_3$ depletion events that occur in the high latitudes.

Mace Head, Ireland (panel b of figure 16) observes air masses that inflow into Europe from the North Atlantic. The model predictions remain within the observed range, however, monthly mean RMSE increases by 38% (from 3.9 ppbv to 5.4 ppbv).

Model error in surface $O_3$ at remote tropical locations, such as Cape Verde Atmospheric Observatory (CVAO) in Cabo Verde (panel c of figure 16), is generally low (2.2 ppbv). The decrease in inorganic iodine emissions from the ocean surrounding these islands increases this error (3.5ppbv, +59%), however, like Mace Head, the model remains within the observed range with the increase in error most notable during spring.

Comparisons between GEOS-Chem and $O_3$ observations in the Antarctic and Southern Ocean have consistently shown a low bias in the model (Young et al., 2013; Sherwen et al., 2016; Schmidt et al., 2016; Pound et al., 2020). As with northern high latitudes, the model is also unable to replicate halogen-driven $O_3$ depletion events which occur during Antarctic spring. The decrease in surface $O_3$ concentrations over the Southern Ocean and Antarctic caused by the increase in Southern Ocean

inorganic iodine emissions exacerbates the underestimate of $O_3$ observations made at Neumayer and Cape Grim (panels d and f of figure 16), increasing RMSE by 56% (from 4.5 ppbv to 7.0 ppbv) and 83% (from 1.8 ppbv to 3.3 ppbv) respectively. The third Southern hemisphere location of Ushaia (panel e of figure 16), which has a small increase in model error (5%, from 2.4 ppbv to 2.5 ppbv).The large increase in the error of the Southern Ocean, particularly Antarctic, surface $O_3$ with still large underestimates in surface IO further indicates missing processes in our understanding of Antarctic $O_3$.

While there are times of better or worse agreement between the model and observations at all locations presented in figure 16, model failure is likely not strongly influenced by year-to-year variability in the iodine emissions. Overall, uncertainties in the chemistry, transport and deposition of iodine species, together with errors and uncertainties in the emission of other species ($NO_x$, VOC's, halogens, particulate) will combine to provide the overall error profile. There is negligible change in area-weighted mean tropospheric OH concentration and tropospheric $CH_4$ lifetime (both <0.2%).

## 11  Conclusions

Here we present a new SML box model that incorporates recent advancements in inorganic iodine chemistry, $O_3$ deposition velocity calculation and improvements to the representation of surface ocean mixing. One key difference between this and previous work is the simulation of depletion of $I^-$ in the SML, which is dependent on the turbulence and the $O_3$ concentration and has been previously observed in experiments using artificial seawater. This results in a dampening of iodine emissions in low wind speed conditions.

From this new box model, we derive parameterised equations for HOI and $I_2$ emissions which can then be implemented in global models. Using these updated equations in GEOS-Chem combined with the most accurate iodide climatology currently available results in a large increase in total global inorganic iodine emissions (+49%) and a small decrease in modelled tropospheric $O_3$ burden (-1.5%). However, it does result in some local reductions in inorganic iodine emissions in equatorial waters and substantially increased emissions from high-latitude waters.

There are still several uncertainties that remain in oceanic iodine chemistry, atmospheric iodine chemistry, and the emissions of iodine from the SML that have not been addressed by this work. In particular, the model does not account for organic-iodine or organic-ozone interactions in the SML or surfactants suppressing ocean-atmosphere exchange. These processes are not sufficiently well understood to include in models but should be a focus of future work.

*Code availability.* GEOS-Chem source code is openly available on GitHub (https://github.com/geoschem/geos-chem). This work used model version 14.1.1 (GCC14.1.1, 2023).

The box model developed here has been made publicly available on GitHub (https://github.com/r-pound/COAGEM) as version 1.1.0 (Pound et al., 2024)

*Data availability.* The complete results for sensitivity runs used to produce the parameterised HOI and $I_2$ have been archived and are openly available (Pound et al., 2023a)

*Author contributions.* R.J.P performed model development, conducted the simulations and analysed the output. L.V.B contributed to model development and analysis. M.J.E., L.J.C. and R.J.P developed the project. All authors contributed to the writing of this manuscript

*Competing interests.* The authors declare that they have no conflict of interest.

*Acknowledgements.* We thank all the reviewers who have provided constructive comments during the review process.

We thank WMO GAW and the individual sites that make up this network, for the availability of the surface ozone data through EBAS, managed by the Norwegian Institute for Air Research.

We thank Hisahiro Takashima for sharing IO observations. The Viking cluster was used during this project, which is a high performance compute facility provided by the University of York. We are grateful for computational support from the University of York, IT Services and the Research IT team

L.J.C, R.J.P and L.V.B acknowledge funding from the European Research Council (ERC) under the European Union's Horizon 2020 programme (Grant agreement No. 833290).

MJE thanks the UK National Centre for Atmospheric Science for funding.

We thank the GEOS-Chem community for developing the model over the past decades.

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

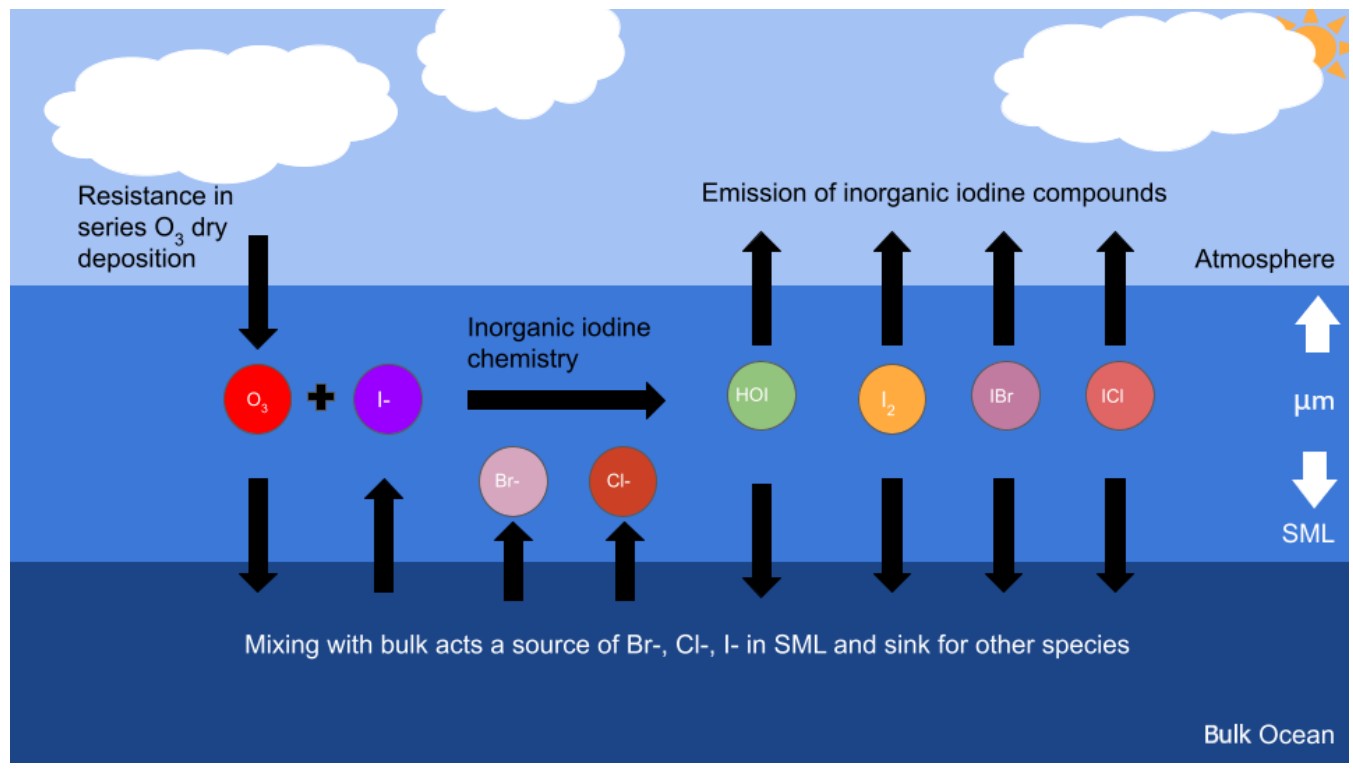

**Figure 1.** Overview diagram describing the physical arrangement of the ocean surface microlayer and the key chemical species included in the model The black arrows represent the chemical fluxes and their net direction in this model.

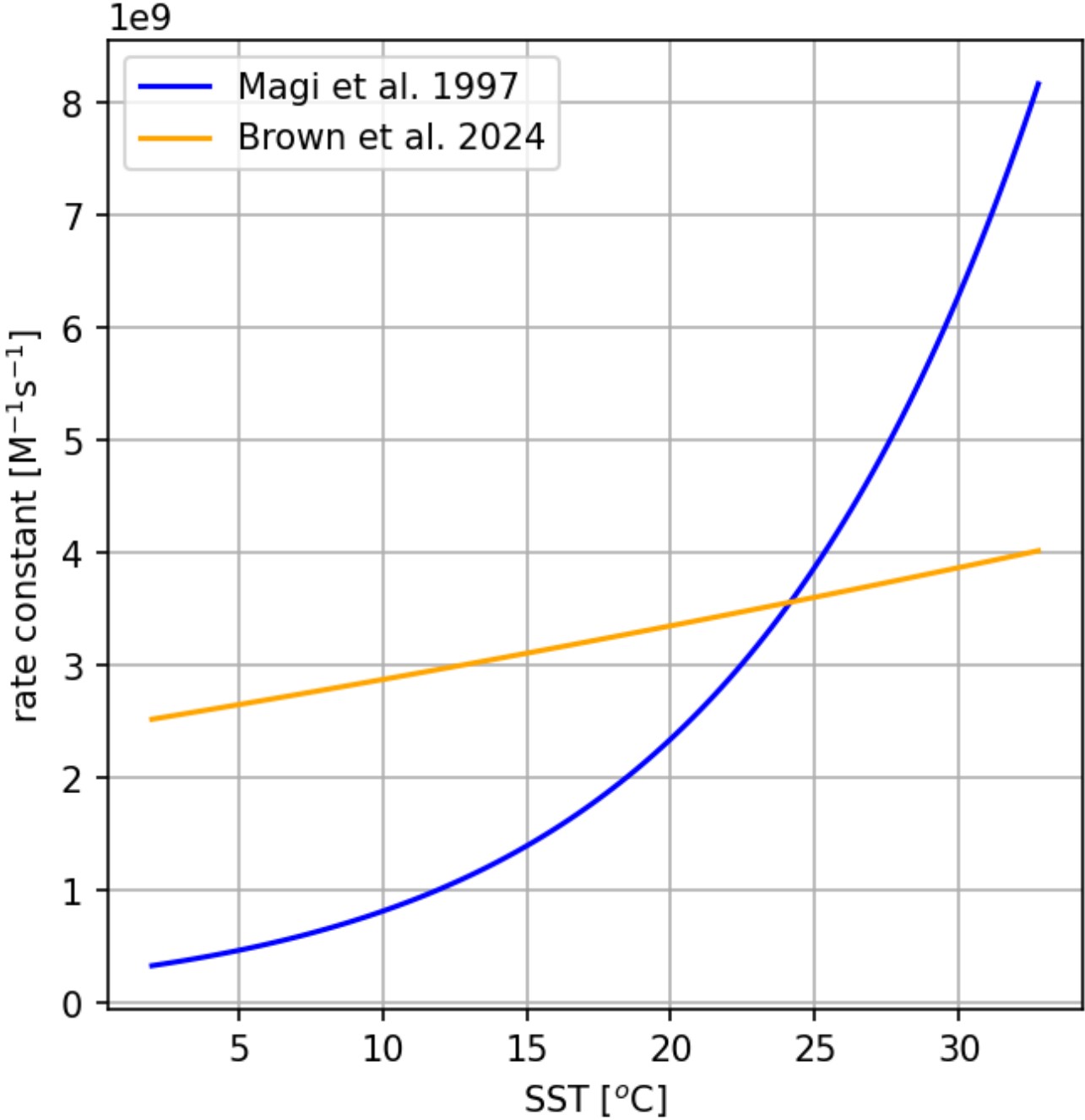

**Figure 2.** Comparison of the two published temperature-dependent rate coefficients from Magi et al. (1997) (blue) and Brown et al. (2024) (orange)

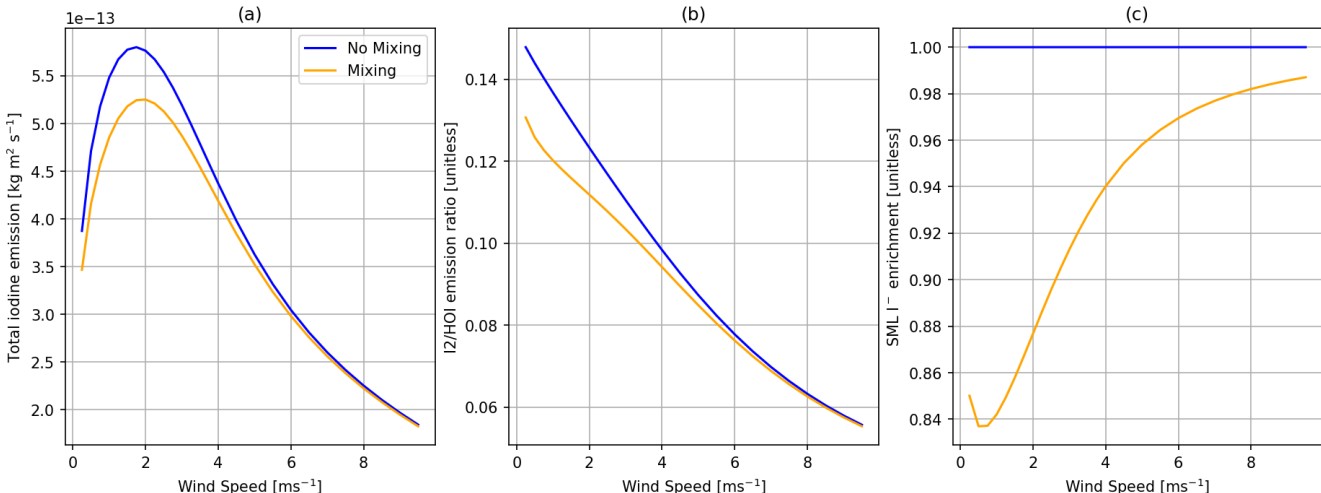

**Figure 3.** Comparisons of the SML model predictions with no mixing of $I^-$ (100 nM in both the SML and bulk layers) (blue) and mixing of $I^-$ from the bulk layer and varying in the SML (orange). a) Total inorganic iodine emission (HOI+$I_2$+IBr+ICl) vs wind speed, b) The ratio of $I_2$/HOI emission vs wind speed and c) SML $I^-$ enrichment (SML concentration/bulk concentration) vs wind speed. All calculations are performed at 30 ppb of atmospheric $O_3$, 100 nM $I^-$ concentration in bulk water, and 285K sea surface temperature (SST).

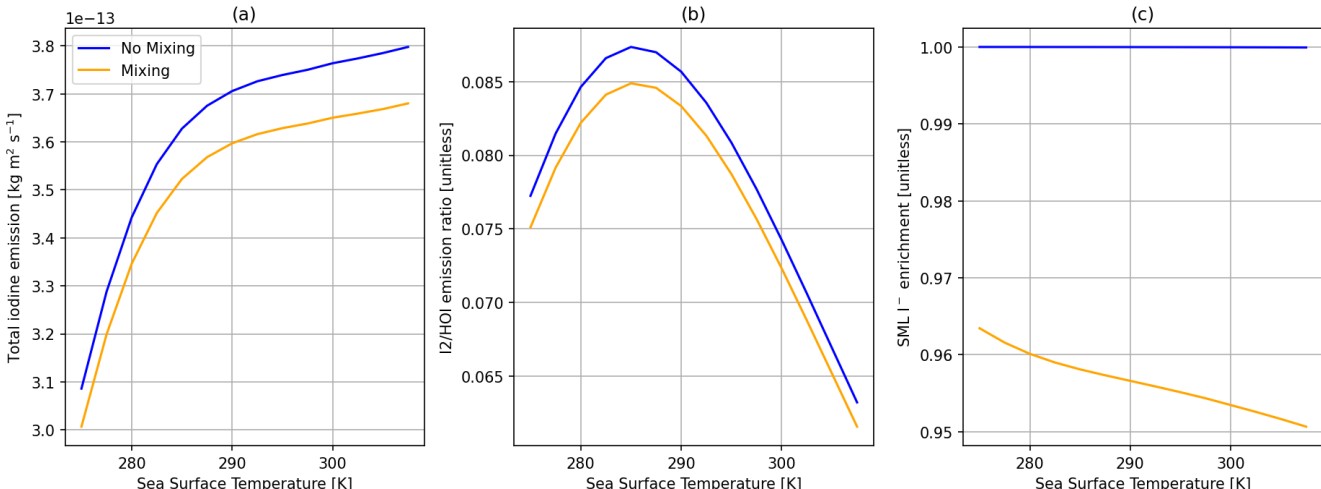

**Figure 4.** Comparisons of the SML model predictions with no mixing of $I^-$ (100 nM in both the SML and bulk layers) (blue) and mixing of $I^-$ from the bulk layer and varying concentration in the SML (orange). a) Total inorganic iodine emission (HOI+$I_2$+IBr+ICl) vs SST, b) The ratio of $I_2$/HOI emission vs SST and c) SML $I^-$ enrichment (SML concentration/bulk concentration) vs SST. All calculated at 30 ppb of atmospheric $O_3$, 100 nM $I^-$ concentration in bulk water, and 5 $ms^{-1}$ wind speed.

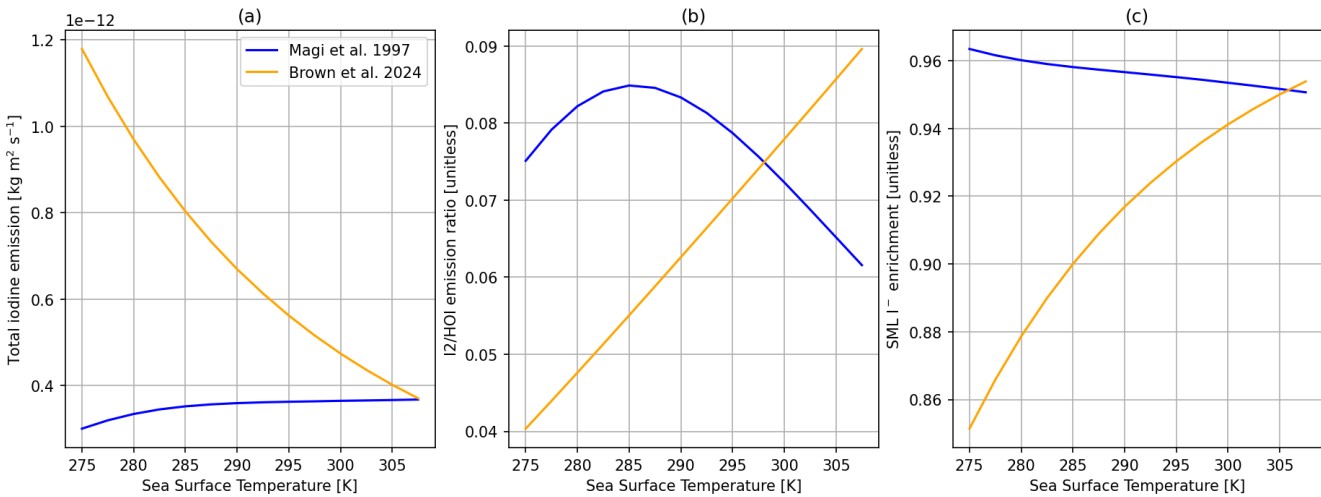

**Figure 5.** Comparisons of the SML model with the $I^- + O_3$ rate reported by Magi et al. (1997) (blue) and Brown et al. (2024) (orange). a) Total inorganic iodine emission vs SST, b) the ratio of $I_2$/HOI emission vs SST and c) SML I- enrichment (SML concentration / bulk concentration) vs SST. All are calculated at 30 ppb of atmospheric $O_3$, 5m/s wind speed, and 100 nM $I^-$ concentration in bulk water.

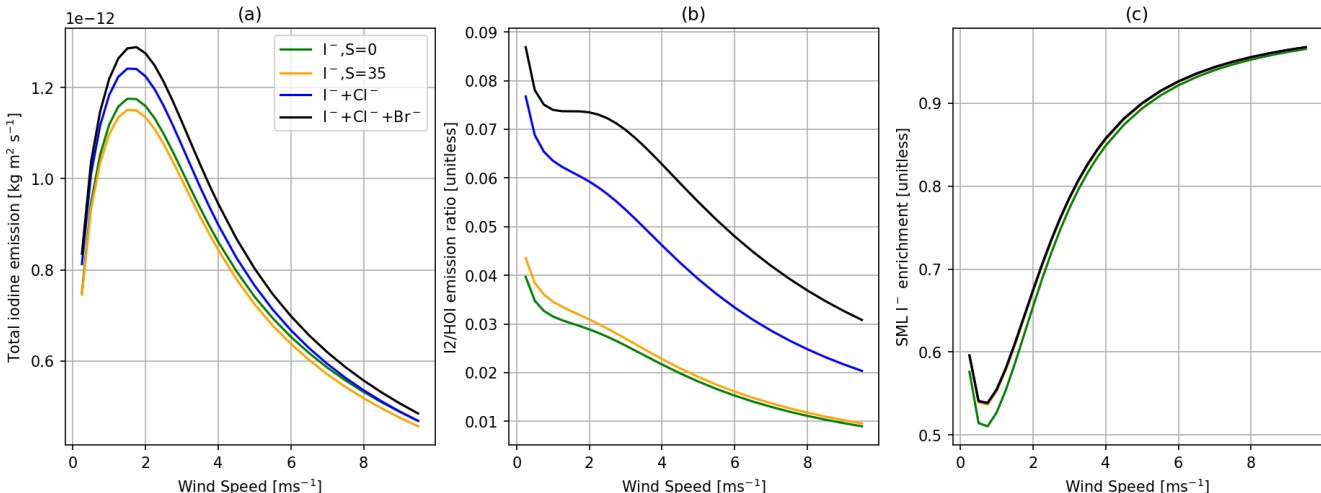

**Figure 6.** Comparisons of the SML model with only iodine chemistry (green), only iodine chemistry but with a salinity of 35 PSU (orange), iodine and chlorine chemistry (blue), and with the full chemistry scheme present (iodine, bromine and chlorine chemistry, black). a) shows total inorganic iodine emission vs ws, b) ratio of $I_2$/HOI emission vs ws and c) SML I- enrichment (SML concentration/bulk concentration) vs ws. All calculated at 30 ppb of atmospheric $O_3$, 285 K SST, and 100 nM $I^-$ concentration in bulk water.

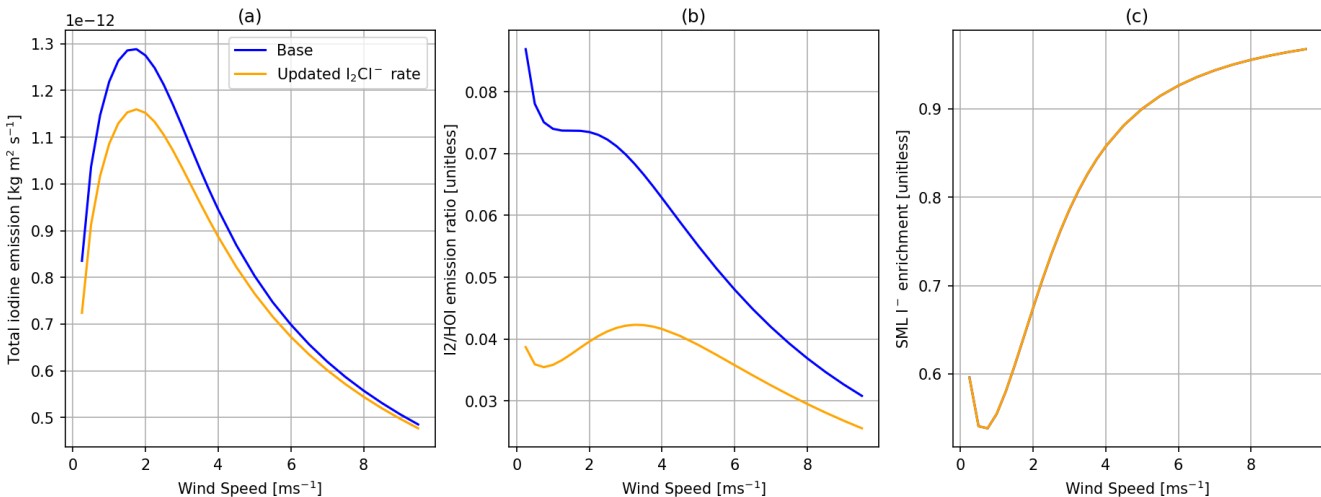

**Figure 7.** Comparisons of the SML model with $I^- + O_3$ rate using the standard chemistry scheme (blue) and the updated equilibrium from Schneider et al. (2023) (orange). a) shows total inorganic iodine emission vs ws, b) ratio of $I_2$/HOI emission vs ws and c) SML I- enrichment (SML concentration/bulk concentration) vs ws all using 30 ppbv of atmospheric $O_3$, 285 K SST, and 100 nM $I^-$.

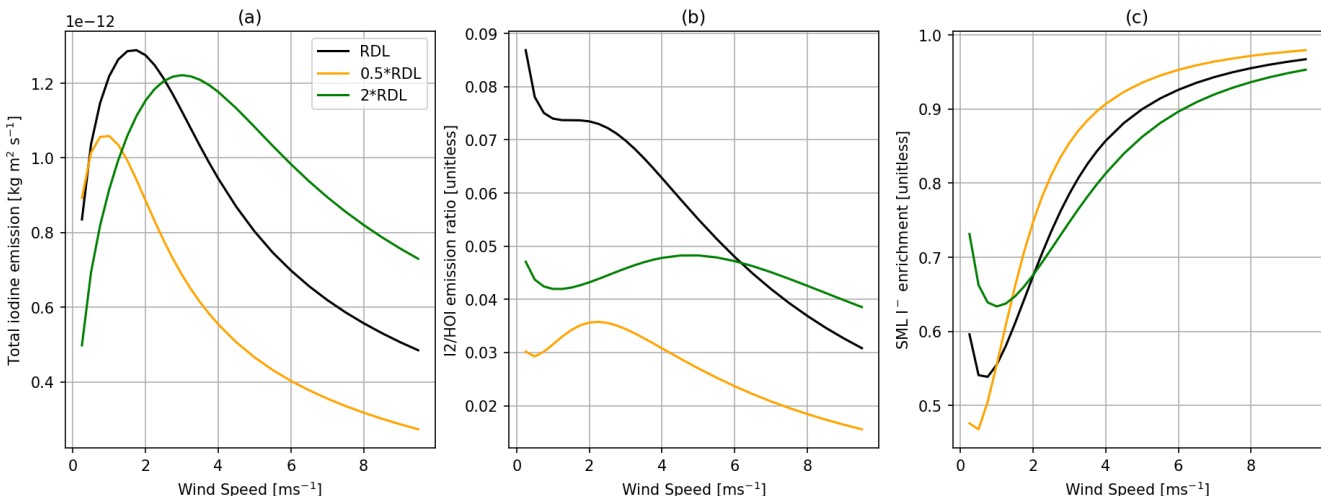

**Figure 8.** Comparisons of the SML model with the depth of the model being half (orange), and twice (green) the reaction-diffusive length (RDL) and the base model using a single reaction-diffusion length (black). a) shows total inorganic iodine emission vs wind speed, b) ratio of I$_2$/HOI emission vs wind speed and c) SML I- enrichment (SML concentration/bulk concentration) vs wind speed all using 30 ppbv of atmospheric O$_3$, 285 K wind speed, and 100 nM I$^-$.

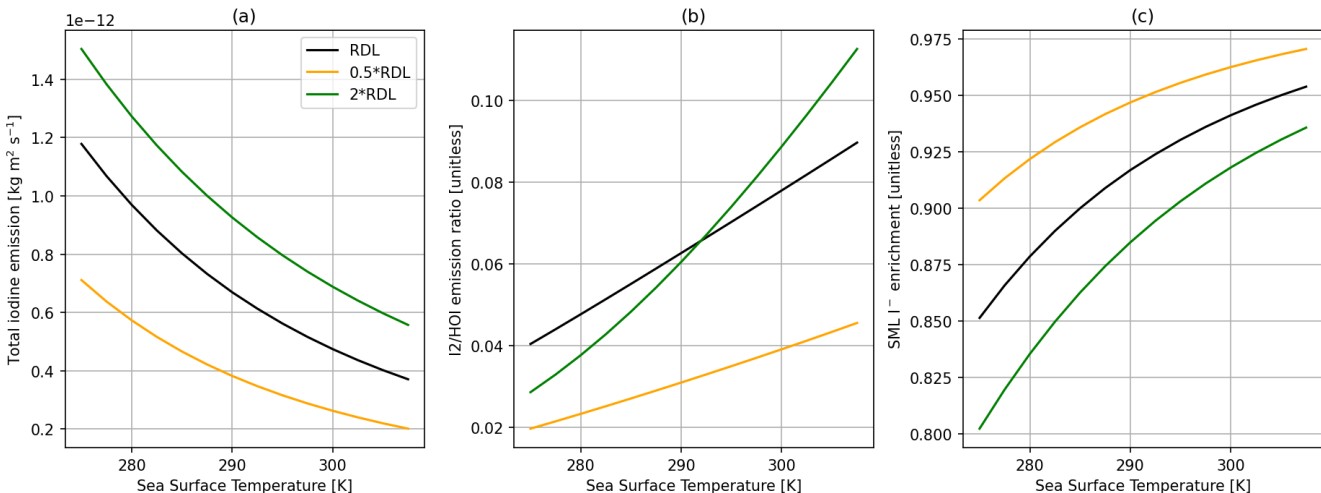

**Figure 9.** Comparisons of the SML model with the depth of the model being half (orange), and twice (green) the reaction-diffusive length (RDL) and the base model using a single reaction-diffusion length (black). a) shows total inorganic iodine emission vs SST, b) ratio of $I_2$/HOI emission vs SST and c) SML I- enrichment (SML concentration/bulk concentration) vs SST all using 30 ppbv of atmospheric $O_3$, 5m/s wind speed, and 100 nM $I^-$.

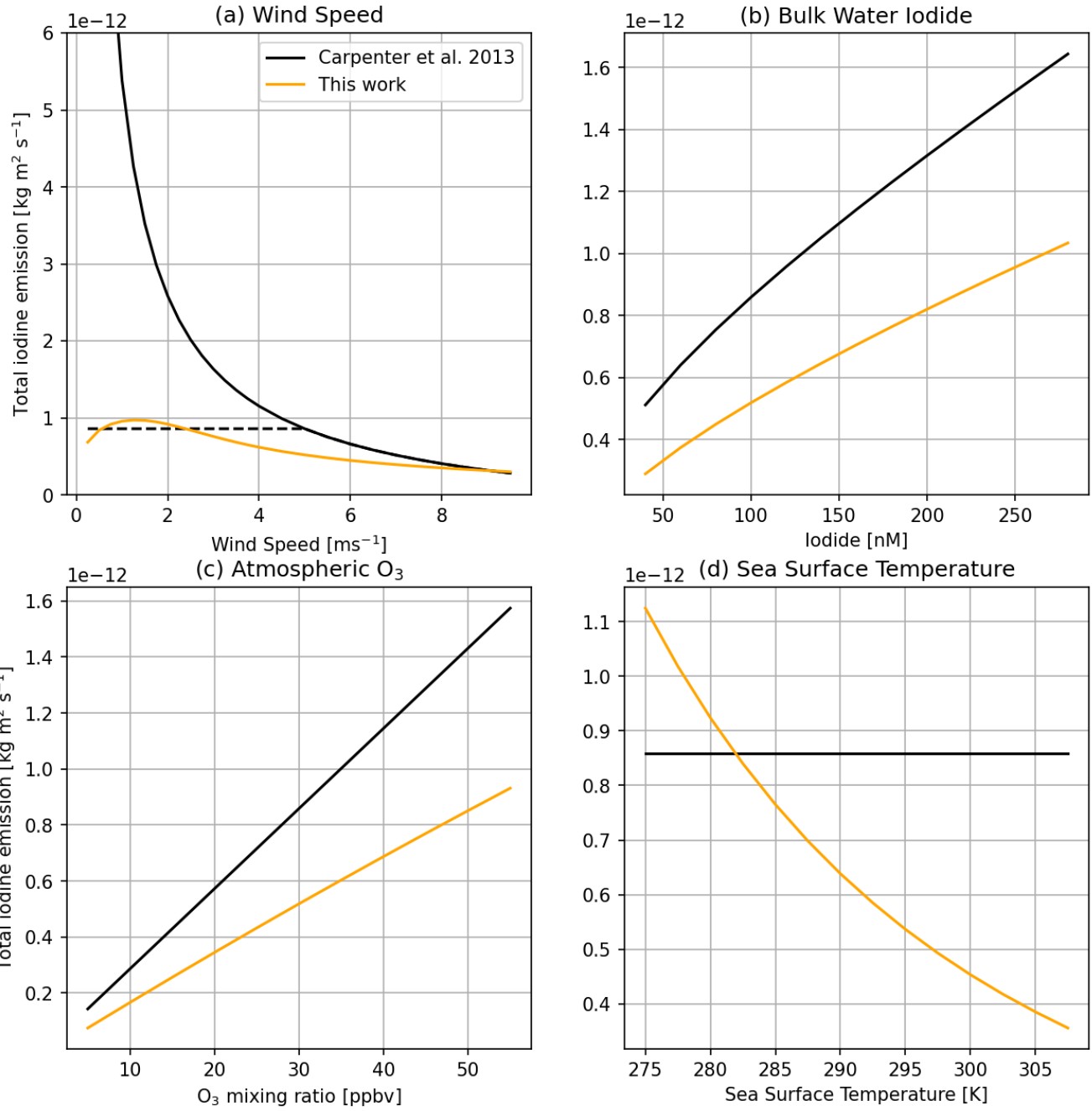

**Figure 10.** Comparison between total iodine emissions from this work and the model as implemented by Carpenter et al. (2013) across a range of wind speeds (a) (both with (dashed black line) and without (solid black line) a minimum wind speed of 5.5 ms$^{-1}$), bulk water iodide concentrations (b), atmospheric $O_3$ mixing ratios (c) and sea surface temperatures (d)

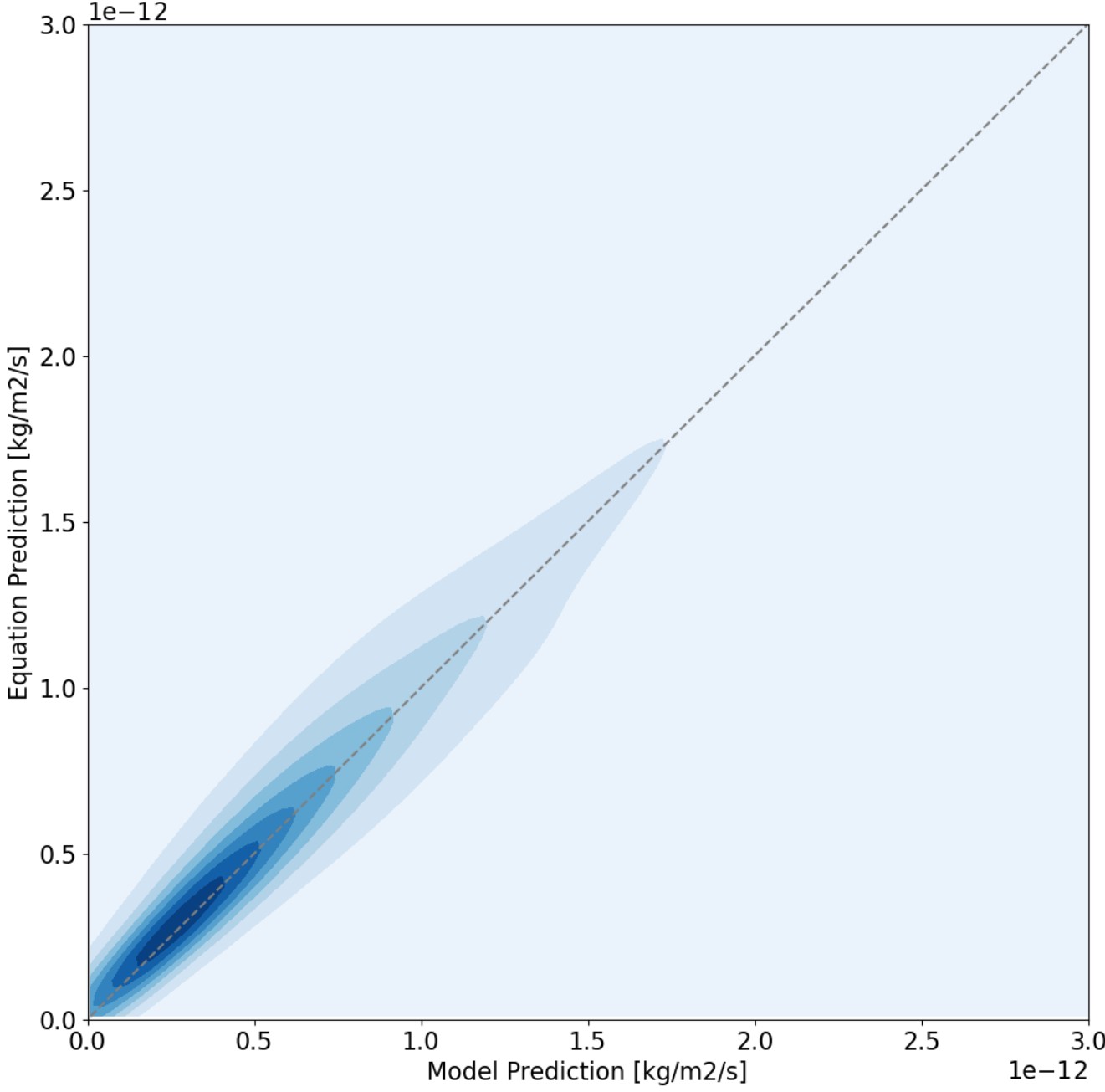

**Figure 11.** Correlation between modelled total inorganic iodine emission and the sum of HOI + $I_2$ predicted using equations 18 and 19

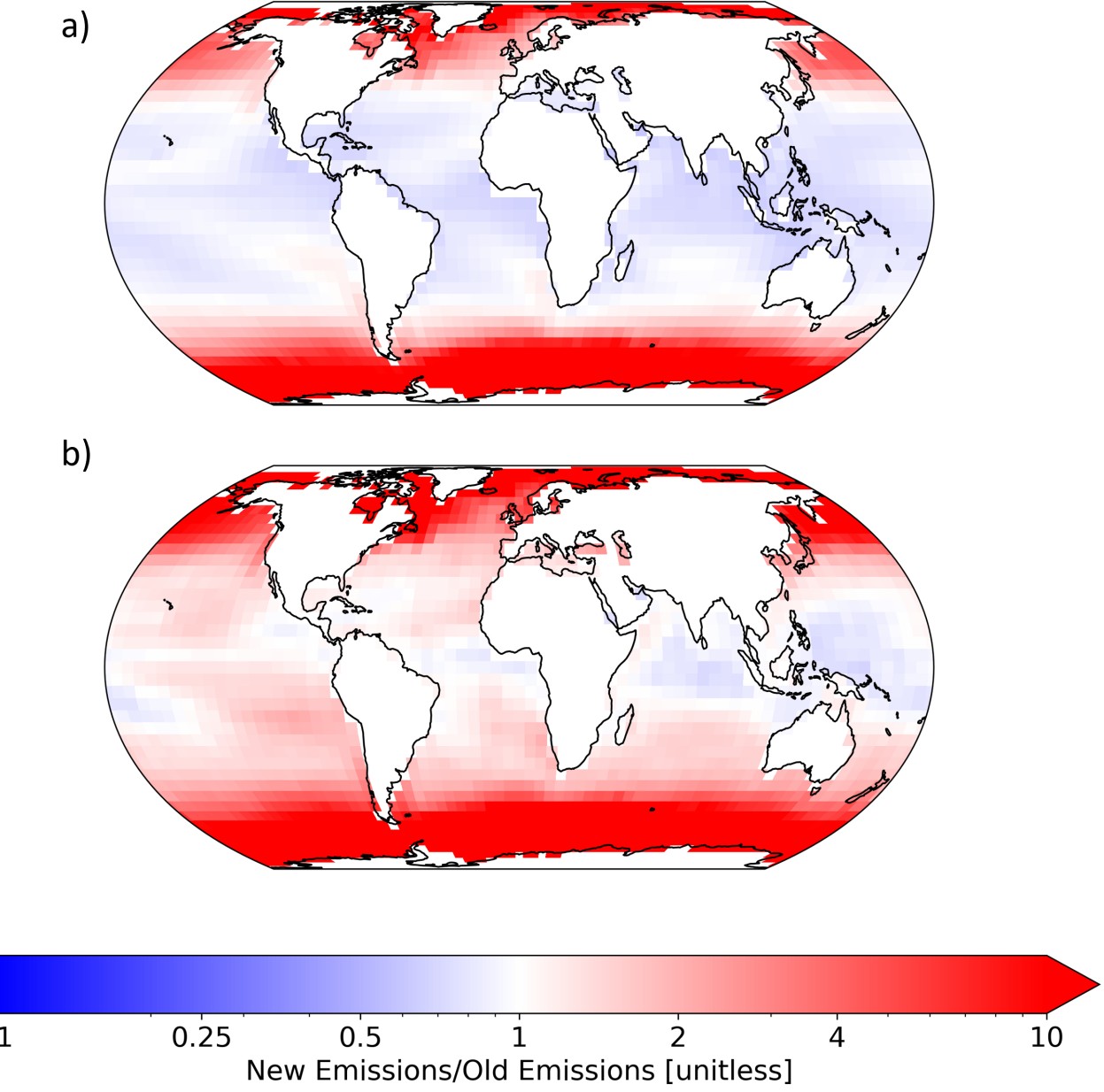

**Figure 12.** Fractional change in the annual mean total inorganic iodine emissions from Carpenter et al. (2013) equations and MacDonald et al. (2014) (base) I⁻ to the new HOI and $I_2$ emission equations (equations 18 and 19) and MacDonald et al. (2014) in panel a. Panel b shows the change from base to the new HOI and $I_2$ emission equations and Sherwen et al. (2019) I⁻. The new version of inorganic iodine emission equations combined with Sherwen et al. (2019) sea surface iodide predicts higher emissions at higher latitudes and a decrease in emissions from warmer, tropical waters.

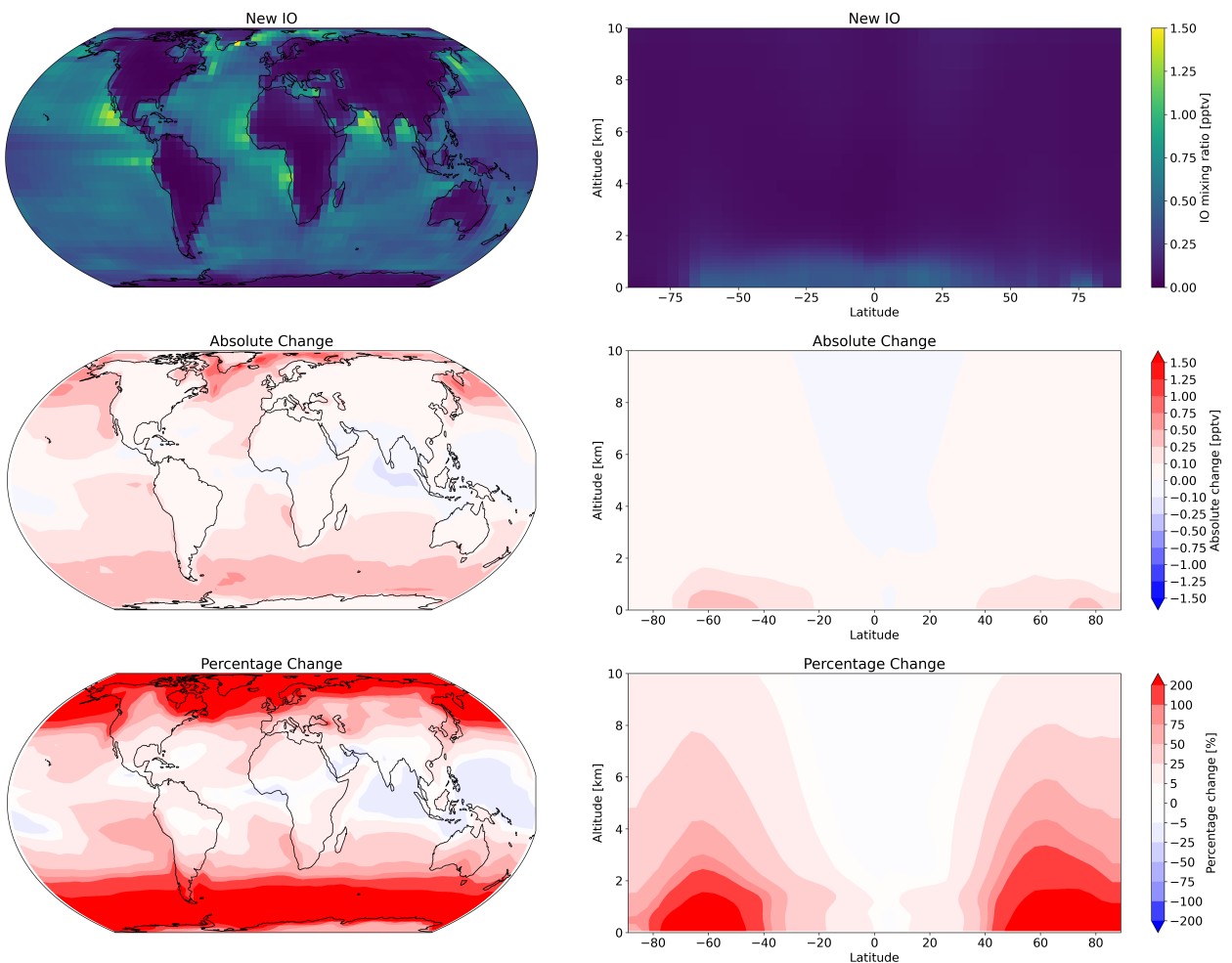

**Figure 13.** Annual mean mixing ratio of IO (top) using new inorganic iodine emissions, absolute change (middle) and percentage change (bottom) in the annual mean atmospheric IO from implementing the new inorganic iodine emissions relative to the "base" model case which used oceanic iodine emissions from Carpenter et al. (2013) and iodide from MacDonald et al. (2014).

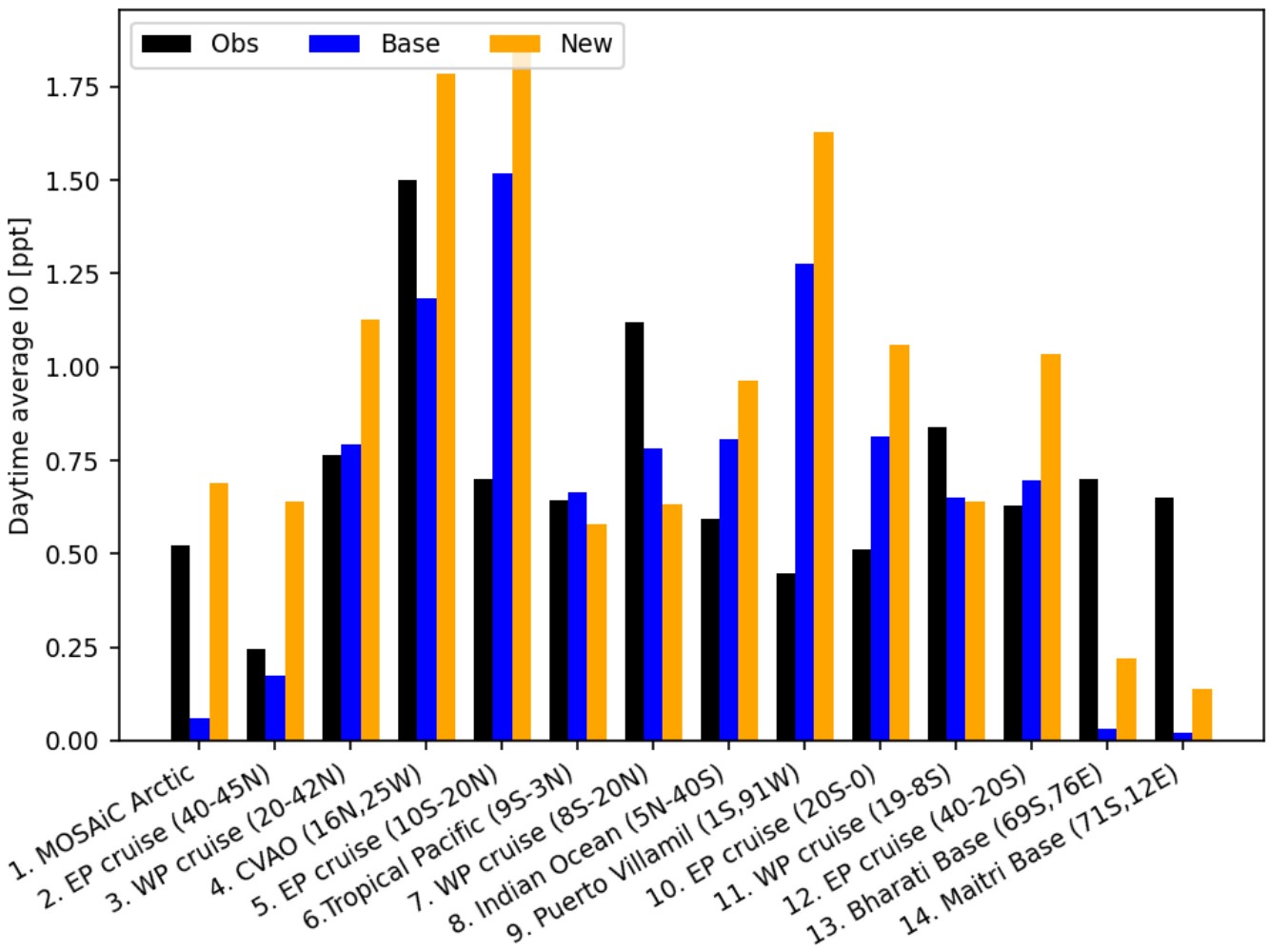

**Figure 14.** Daytime surface average IO mixing ratio from coastal sites and ocean cruises with observations (black) from reporting periods in different years. Model values are monthly mean daytime surface values taken from the same reporting month and location but from years 2020/21 where the base (blue) uses HOI and $I_2$ emissions from Carpenter et al. (2013) driven by MacDonald et al. (2014) iodide and new (orange) uses the HOI and $I_2$ emissions presented in this work driven by Sherwen et al. (2019) iodide. References: (1) Mahajan (2022), (2, 5, 10, 12) Mahajan et al. (2012), (3, 7, 11) Großmann et al. (2013), (4) Mahajan et al. (2010), (6) Takashima et al. (2022), (8) Mahajan et al. (2019a, b), (9) Gómez Martín et al. (2013), (13, 14) Mahajan et al. (2021)

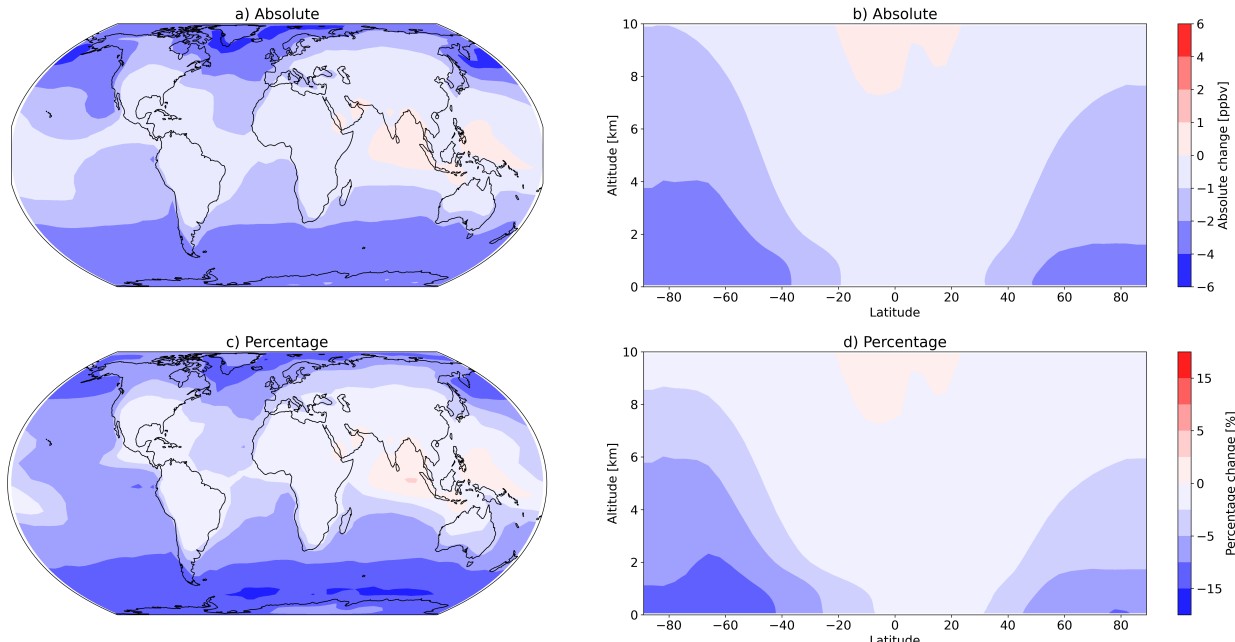

**Figure 15.** Absolute and percentage change in surface $O_3$ (a and c), absolute and percentage change in zonal $O_3$ (b and d) due to changing the Carpenter et al. (2013) inorganic iodine emissions from the ocean to the equations presented in this work. The largest changes occur in the surface levels of the model, with the largest relative decrease in surface $O_3$ occurring over the Southern Ocean and the largest relative increase occurring over the Indian Ocean.

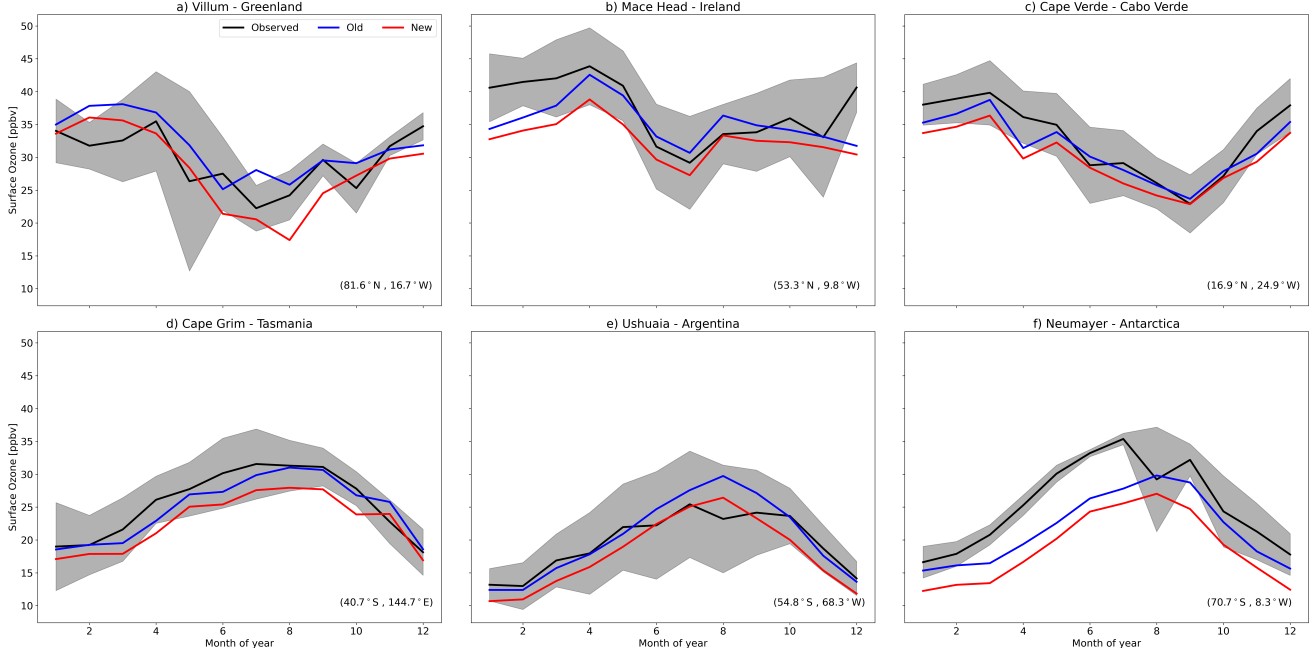

**Figure 16.** Predictions and observations of monthly average surface ozone mixing ratio from the model using the old iodine emissions (old) and the model using equations 18 and 19 (new) for six GAW stations (with the latitude and longitude for each station at the bottom right) with the shaded region representing the $25^{th}$ to $75^{th}$ percentiles. Observational data from 2014.

| Variable | Definition | Constant / Dependance / Input |
|---|---|---|
| $T$ | Sea surface temperature | Input |
| $u_{10}$ | 10 m wind speed | Input |
| $u^*$ | friction velocity | $u_{10}$ |
| $u_w^*$ | waterside friction velocity | $u_{10},u^*$ |
| $C_D$ | drag coefficient | $u_{10}$ |
| $[I^-]$ | iodide concentration | Input |
| $[O_3]$ | ozone concentration | Input |
| $C_b$ | concentration in the bulk ocean | Input |
| $S$ | Salinity | Input |
| $C_a$ | concentration in the air | Input |
| $\kappa$ | von Kármán constant | $\sim 0.4$ |
| $S_{cw}$ | Schmidt number in water | $T,S$ |
| $S_{ca}$ | Schmidt number in air | $T$ |
| $S_{c600}$ | Schmidt number of $CO_2$ at 20 °C | 600 |
| $r_a$ | atmospheric resistance to dry deposition | $u_{10}, u^*$ |
| $r_b$ | quasilaminar sublayer resistance to dry deposition | $u^*,S_{ca}$ |
| $D$ | Diffusivity of $O_3$ in water | $T$ |
| $k$ | second-order rate coefficient of $O_3 + I^-$ | $T$ |
| $a$ | chemical reactivity of $O_3 + I^-$ | $[I^-],k$ |
| $\delta_m$ | depth of SML reaction-diffusion layer | $a,D$ |
| $\alpha$ | solubility of $O_3$ in water | $T$ |
| $r_c$ | surface resistance to dry deposition | $a,D,\delta_m,\kappa,u_w^*,\alpha$ |
| $v_d$ | dry deposition velocity | $r_a,r_b,r_c$ |
| $H$ | unitless Henry's law | $T,S$ |
| $k_a$ | air-side transfer velocity | $S_{ca},\kappa,u^*,C_D$ |
| $k_w$ | water-side transfer velocity | $u_{10},S_{cw},S_{c600}$ |
| $R$ | surfactant scale factor | 0.9 |
| $F_a$ | Net flux from the SML to the atmosphere | $k_a,H,C_a,C_{sml}$ |
| $F_b$ | Net flux (molecular transfer) from the SML to the bulk ocean | $k_w,R,C_b,C_{sml}$ |
| $F_r$ | Net flux (surface renewal) from the SML to the bulk ocean | $u_{10},C_b,C_{sml}$ |
| $C_{sml}$ | concentration in the SML | $F_a,F_b,F_r$ |

**Table 1.** All input variables and calculated parameters along with their definitions and dependencies used by the model presented in this work for calculating the dry deposition of $O_3$ into the SML and fluxes of inorganic halogens to the atmosphere and bulk ocean from the SML.

Table 2: All reactions included in the chemistry scheme of this SML model with forward and reverse rate constants (where applicable) and accompanying references. Numbered reactions with a and b denote different rates explored in the sensitivity analysis conducted in this paper. (1) A = $1.44 \times 10^{22}$ M$^{-1}$s$^{-1}$, Ea = 73.08 kJ mol$^{-1}$, (2) A = $2.6 \times 10^{11}$ M$^{-1}$s$^{-1}$, Ea = 10.6 kJ mol$^{-1}$, [3] assumed reaction based on theoretical calculation

| Number | Reaction | Forward rate | Reverse rate | Reference |
|--------|----------|--------------|--------------|-----------|
| R1a | $O_3 + I^- \rightarrow$ IO- | (1) | NA | Magi et al. (1997) |
| R1b | | (2) | NA | Brown et al. (2024) |
| R2 | $I_2 \leftrightarrow I_2OH^- + H^+$ | 3.2 | $2.0 \times 10^{10}$ | Lengyel et al. (1993) |
| R3 | $I_2OH^- \leftrightarrow HOI + I^-$ | $1.34 \times 10^6$ | $4.0 \times 10^8$ | Lengyel et al. (1993) |
| R4 | $I^- + I_2 \leftrightarrow I_3^-$ | $6.2 \times 10^9$ | $8.9 \times 10^6$ | Lengyel et al. (1993) |
| R5 | $HOI + HOI \leftrightarrow H^+ + I^- + HIO_2$ | 25 | $2.0 \times 10^{10}$ | Paquette (1989) |
| R6 | $I_2 + OH^- \leftrightarrow HOI + I^-$ | $7.0 \times 10^4$ | $2.1 \times 10^3$ | Sebők-Nagy and Körtvélyesi (2004) |
| R7 | $HOI \leftrightarrow IO^- + H^+$ | 0.1 | $1 \times 10^{10}$ | Paquette (1989) |
| R8 | $HOI + IO^- \rightarrow HIO_2 + I^-$ | 15 | NA | Bichsel and von Gunten (2000) |
| R9 | $HIO_2 + HOI \leftrightarrow IO_3^- + I^- + 2H^+$ | 240 | $1.2 \times 10^3$ | Paquette (1989) |
| R10 | $H_2OI^+ \leftrightarrow HOI + H^+$ | $9.0 \times 10^8$ | $2.0 \times 10^{10}$ | Lengyel et al. (1993) |
| R11 | $I_2 + H_2O \leftrightarrow H_2OI^+ + I^-$ | 0.12 | $1.0 \times 10^{10}$ | Lengyel et al. (1993) |
| R12 | $HOI + Br^- + H^+ \leftrightarrow IBr$ | $4.1 \times 10^{12}$ | $8.0 \times 10^5$ | De Barros Faria et al. (1993) |
| R13 | $HOI + Cl^- + H^+ \leftrightarrow ICl$ | $2.9 \times 10^{10}$ | $2.4 \times 10^6$ | Wang et al. (1989) |
| R14 | $I_2 + Br^- \leftrightarrow I^- + IBr$ | $4.64 \times 10^3$ | $2.0 \times 10^9$ | De Barros Faria et al. (1993) |
| R15a | $I_2 + Cl^- \leftrightarrow I_2Cl^-$ | $8.33 \times 10^4$ | $5.0 \times 10^4$ | Kumar et al. (1986) |
| R15b | | $8.33 \times 10^4$ | $5.0 \times 10^3$ | Schneider et al. (2023) |
| R16 | $ICl_2^- \leftrightarrow ICl + Cl^-$ | $1.1 \times 10^9$ | 1.5 | Kumar et al. (1986) |
| R17 | $I^- + ICl \leftrightarrow I_2Cl^-$ | $1.1 \times 10^9$ | 1.5 | Kumar et al. (1986) |
| R18[3] | $ICl_2^- + I^- \rightarrow I_2Cl^- + Cl^-$ | $1.0 \times 10^6$ | NA | Kumar et al. (1986) |
| R19 | $HOCl + I^- + H^+ \rightarrow ICl + H_2O$ | $3.5 \times 10^{11}$ | NA | Nagy et al. (1988) |
| R20 | $HOI + HOCl \rightarrow HIO_2 + Cl^- + H^+$ | $5.0 \times 10^5$ | NA | Citri and Epstein (1988) |
| R21 | $HIO_2 + HOCl \rightarrow IO_3^- + Cl^- + 2H^+$ | $1.5 \times 10^3$ | NA | Lengyel et al. (1996) |

| Number | Reaction | Forward $k$ | Reverse $k$ | Reference |
|---|---|---|---|---|
| | | Table 2 –*Continued from previous page* | | |
| R22 | $Cl^- + O_3 + H^+ \rightarrow HOCl + O_2$ | $1.1 \times 10^5$ | NA | Levanov et al. (2019) |
| R23 | $Br^- + O_3 + H^+ \rightarrow HOBr + O_2$ | 11.7 | NA | Haag and Hoigné (1983) |
| R24 | $HOBr + Cl^- + H^+ \rightarrow BrCl + H_2O$ | $5.6 \times 10^9$ | NA | Wang et al. (1994) |
| R25 | $HOBr + Br^- + H^+ \rightarrow Br_2 + H_2O$ | $1.6 \times 10^{10}$ | NA | Beckwith et al. (1996) |
| R26 | $HOCl + Cl^- + H^+ \rightarrow Cl_2 + H_2O$ | $2.2 \times 10^4$ | NA | Wang and Margerum (1994) |
| R27 | $HOCl + Br^- + H^+ \rightarrow BrCl + H_2O$ | $1.3 \times 10^6$ | NA | Kumar and Margerum (1987) |
| R28 | $BrCl + H_2O \rightarrow HOBr + Cl^- + H^+$ | $1.0 \times 10^5$ | NA | Wang et al. (1994) |
| R29 | $Br_2 + H_2O \rightarrow HOBr + Br^- + H^+$ | 97 | NA | Beckwith et al. (1996) |
| R30 | $Cl_2 + H_2O \rightarrow HOCl + Cl^- + H^+$ | 22 | NA | Wang and Margerum (1994) |
| R31[3] | $BrCl + Br^- \rightarrow Br_2Cl^-$ | $5.0 \times 10^9$ | NA | Michalowski et al. (2000) |
| R32[3] | $Br_2 + Cl^- \rightarrow Br_2Cl^-$ | $5.0 \times 10^9$ | NA | Michalowski et al. (2000) |
| R33[3] | $BrCl + Cl^- \rightarrow BrCl_2^-$ | $5.0 \times 10^9$ | NA | Michalowski et al. (2000) |
| R34 | $Br_2Cl^- \rightarrow Br_2 + Cl^-$ | $3.9 \times 10^9$ | NA | Wang et al. (1994) |
| R35 | $Br_2Cl^- \rightarrow BrCl + Br^-$ | $2.8 \times 10^8$ | NA | Wang et al. (1994) |
| R36 | $BrCl_2^- \rightarrow Cl_2 + Br^-$ | 690 | NA | Wang et al. (1994) |

**Table 3.** Comparison of $I_2$ emissions from published experimental studies with the SML model of this study, run using the experiment parameters. Ranges of $I_2$ emissions represent the range of both measured and calculated flux from the range of experimental inputs used. Model results were obtained with R=1, as no organics are present in these experimental results. a) pH 8 seawater spiked with iodide. b) pH 8 buffered solution with 0.5 M chloride and $1 \times 10^{-6}$ M iodide. c) Artificial seawater containing iodide, bromide and chloride, buffered to pH 8. d) Iodide only in buffered pH 8. solution

| Study | $O_3$ [ppbv] | Iodide [nM] | Stirred | Temperature [°C] | $I_2$ emission [molecules cm$^{-2}$ s$-1$] | Model prediction [molecules cm$^{-2}$ s$-1$] |
|---|---|---|---|---|---|---|
| Carpenter et al. (2013) [a] | 70 | 10000-30000 | Yes | 18 | $0.6$-$1.8 \times 10^{11}$ | $0.2$-$0.3 \times 10^{11}$ |
| MacDonald et al. (2014) [b] | 222 | 1000 | No | 3-25 | $13 \pm 4 \times 10^9$ | $0.3$-$1.1 \times 10^9$ |
| Tinel et al. (2020) [c] | 20-110 | 1200 | Yes | 17 | $3$-$10 \times 10^8$ | $7.3$-$40 \times 10^8$ |
| Tinel et al. (2020) [c] | 34.7 | 400-10000 | Yes | 17 | $2$-$100 \times 10^8$ | $2.9$-$78 \times 10^8$ |
| Schneider et al. (2023) [d] | 95-110 | 390 | No | 22-25 | $7.7 \times 10^9$ | $0.2$-$0.6 \times 10^9$ |