# Peer review of "An improved estimate of inorganic iodine emissions from the ocean using a coupled surface microlayer box model"

_EGUsphere, 2023_

## Author Comment (AC1)

The authors would like to thank the reviewers for their comments and suggestions. As well as the changes to the paper detailed in the following responses, the results shown in figures 2- 7 and 9 have been updated to reflect the correction to the rate coefficient from Brown et.al 2024 (previously Brown et.al 2023 under review). We have also updated equations 18 and 19 for predicting HOI and I2 emissions and present updated results for the global modelling runs as well as discussing the sensitivity to the iodide field used. All page and line numbers given in the response to reviewer comments are in reference to the tracked changes version of the manuscript.

**Reviewer 1**

In this manuscript, Pound et al. present a fairly comprehensive box model of the ocean surface microlayer (SML), coupling the deposition of ozone with the chemistry in the SML, with a major focus on oceanic emissions of inorganic iodine. This is great effort, providing a mechanistic representation of air-SML-ocean processes for chemistry-climate models, and the model can (in theory) be expanded for other compounds of interest. The manuscript is generally well prepared. I recommend this manuscript for publication in EGUsphere after the following comments and concerns are addressed.

My main concern is the model representation of SML itself. The thickness of the SML is operationally defined, with commonly quoted thickness ranging between 1 micron to up to 1 millimeter. The thickness and composition of SML may also depend on wind, although the dependency remains unclear. It remains unclear how the SML thickness affects the mass transfer as well as the subsequent ozone deposition and iodine emission. I acknowledge that there are many aspects remain poorly understood to include in the model but the impact of SML thickness looks like a lower hanging fruit to me.

- The authors thank the reviewer for these comments and have clarified that the model considers only the reaction-diffusion length of ozone as this is where the reaction with $O_3$ + $I^-$ primarily occurs (lines 63 and 118-122). The "SML" is a generic term which has different meanings in different contexts, whereas the reaction-diffusion length of ozone is defined by equation 8. This depth has a direct dependence on temperature and an indirect dependence on wind speed and $O_3$ concentration via [$I^-$]. In low turbulence or high atmospheric $O_3$ conditions, [$I^-$] becomes depleted and increases the depth of the reaction-diffusion layer.
- We have also included an additional discussion of the sensitivity of the predicted inorganic iodine emission to the depth of the reaction-diffusion layer in a new section of discussion (Section 6, lines 288-296, and figures 8 and 9)

What's also unclear is the scalability of the SML model. Properties of SML have mostly been derived using coastal/offshore/nearshore observations, and observations obtained in the open ocean are not limited. Any robust measurements supporting the widespread existence of SML? I understand this is perhaps beyond the scope of this work. But it would be tremendously helpful for audience who are not entirely familiar with this topic if

the authors can include a brief summary on this, which certainly puts this work into a broader perspective

- The authors have included additional discussion of the existence of the SML across the oceans (Line 59-61) and additional perspective on the portion of the SML where the reaction between $O_3$ and I- occurs.

    "*The sea surface microlayer (SML) covers the world's oceans to a significant extent, ranging in depth from 1-1000 μm and having distinct chemical and biological properties from underlying waters, and is the interface between the ocean and atmosphere (Wurl et al., 2011; Cunliffe et al., 2013; Wurl et al., 2017)*"

Minor/technical comments:

Page 2 Line 38: "Recent work has also supports atmospheric iodine playing an important role in *partial* formation in the MBL…" Should this be *particle*?

- The text of line 42 has been updated to now correctly read

    "*Recent work has also supports atmospheric iodine playing an important role in particle formation in the MBL…*"

Page 2 Line 40: I don't think the in-text citation format is correct.

- The citation for Tham et.al 2021 has been updated to be in parentheses

Page 3 Lines 86-88: this is where the thickness of SML may play a role. Thickness and the reacto-diffuso-length together determine if the reaction occurs at the surface or the bulk. I would imagine the thickness also affects the depletion and replenishment of iodide in the SML.

- The model only considers the reaction and depletion occurring within the reacto-diffusive length. This has been further clarified in the response to the earlier comment.

Page 5 Lines 128-129: If the aqueous-phase chemistry of ozone (in SML) is explicitly represented in the model already, why would you need this r_c term in the resistance model? Ultimately the tendency of ozone in gas-phase consists of turbulent transport (r_a) to the surface and mass transport across the quasilaminar sublayer (r_b), as well as whatever chemistry occurs in the gas-phase and the aqueous-phase. For simple deposition models without explicit representation of aqueous-phase chemistry, r_c term summarizes the impacts of aqueous-phase chemistry in a simplified psuedu-1st order manner (a = k[I-]). But this work already has detained chemistry, would including r_c term be double-counting?

- The authors thank the reviewer for their comments. This model has been designed to fit into the current framework (resistance-in-series) used in existing

models. The current approach of calculating a deposition velocity including the $r_c$ term does not result in double counting as the flux of $O_3$ entering the ocean surface (using the resistance-in-series dry deposition velocity) equates to the amount of $O_3$ lost by reaction with iodide. The authors do agree that alternative methods of calculating the flux of $O_3$ into the ocean surface are worth exploring, particularly in systems expanded beyond that presented here. Future work will aim to construct the model without a reliance on the surface resistance.

Page 5 Equations 4-6: please make sure all variables and parameters in those equations are clearly defined and explained in the text.

- Page 5 and 6, with the definition of terms from equations 4-6 has been rewritten to collect the definitions of these terms together and improve the clarity of this section.

Page 5 Equation 9: shouldn't this be the diffusivity in the SML? With all the organics and surfactants, would you expect the diffusivity be the same as (dilute) water?

- We have updated the definition of Equation 9 to specify that it is for the diffusivity of $O_3$ in the SML. This equation does not account for the impact of surfactants on the diffusivity of $O_3$, and at this stage, the role of organics has not been considered in this model. This is a limitation of the current model and a focus of future research. There is still a great deal of uncertainty in the role organics combined with iodine play in the dry deposition of $O_3$ to the ocean surface and the subsequent emission of iodine species. Further development of this in the SML model presented here will require the support of additional experiment results.

  The definition of D (line 158-160) has been updated to include the following text

  *"This calculation of D does not account for the impact of organics (particularly surfactants) which will impact the transfer of $O_3$ into the SML, this model is currently limited to inorganic chemistry and the limitations of this are discussed further in the conclusions."*

Page 10 Section 7: please clarify how the modeled ranges are calculated. I understand it is extremely challenging to mimic the experimental conditions with a model like this, therefore the fact that the model captures the orders of magnitudes is impressive in my opinion. I do wonder to what degree would the thickness of the SML affect the modeling results.

- The model ranges presented in Table 3 for Section 7 represent the range of values used in the experimental conditions rather than model uncertainties. The text in the caption for table 3 has been updated to

  *"Comparison of $I_2$ emissions from published experimental studies with the SML model of this study, run using the experiment parameters. **Ranges of***

*I₂ emissions represent the range of both measured and calculated*
*flux from the range of experimental inputs used. Model results .."*

Pages 9-10. The GEOS-Chem implementation is interesting. However, I can't help but notice the modeling period (2020-2021) enters a strong/rare La Niña event (2020-2023). The model-measurement comparison presented in this work obviously ignores inter-annual variabilities. If the global iodide input (Sherwen et al) is a climatology that does not include inter-annual variability, then perhaps there is no point to extend the simulation period to match the measurement years. Please comment if these IO measurements (Figure 12) would be subject to considerable inter-annual variability.

- Sherwen et.al 2019 is a climatology with this now stated in the text. Interannual variability on line 365

  "Here, we use the up-to-date, machine learning-derived, iodide climatology…"

- The interannual variability of IO and any relation with El Nino and La Nina events is beyond the scope of this study but the authors agree that this does warrant future investigation.

**Reviewer 2**

Review of manuscript ID 'egusphere-2023-2447' titled 'An improved estimate of inorganic iodine emissions from the ocean using a coupled surface microlayer box model' by Pound et al.

This manuscript offers a box model of the ocean surface microlayer (SML) where ozone deposition and chemistry lead to the emission of iodine from the sea surface. This model aims to provide a mechanistic representation of air-SML-ocean processes leading to emissions, which have traditionally been parameterised in models. The new emissions of inorganic iodine show a large spatial difference compared to the past emission inventories, but the global mean is similar. However, the comparison with observations does not show an obvious improvement either in terms of iodine or ozone observations.

Overall, the manuscript is well-written, and the model is well-detailed. This represents a step up from the current description in models, and hence, I recommend this manuscript for publication after addressing the following comments:

Comments:

While the study offers new parameterisations for inorganic iodine emissions, the comparison with observations does not show a large improvement. Even if one does not consider the polar regions where the mismatch is largest (attributed possibly to sea-ice emissions), the new parameterization does not significantly improve the comparison.

Indeed, no statistical analysis is provided to quantify whether there is an improvement or not.

- The authors have included additional discussion of model-observation IO comparisons (lines 399-401), including relative mean bias along with the existing discussion of root mean square error both including and excluding polar observations.

    "*The change in iodine emissions has little effect on the average model root mean square error of atmospheric IO, increasing it from 0.48 to 0.62 ppt (0.43 ppt to 0.58 excluding polar observations) with a change in the relative mean bias from -0.43 ppt to 0.43 ppt (-0.07 ppt to 0.55 ppt excluding polar 400 observations), suggesting that there are still uncertainties…*"

This begs the question, what else is affecting the emissions? It would be helpful if there were a section in the discussions on the possible effects of surfactants or other mechanisms that can inhibit emissions (in most cases, the model overestimates the observations). This is mentioned in the conclusions, but no discussion on these effects and studies that have shown them is presented.

- The authors agree that more research is needed to understand the complex relationship between organics and iodine emissions from the ocean. In addition to Lines 70-71 & 74-75 in the Introduction discussing evidence of the role that surfactants play in suppressing iodine emissions, lines 460-463 in the Conclusions acknowledges that the model lacks a description of the role of organics in iodine emissions. We have also included further discussion (lines 401-404) to acknowledge current uncertainties in the iodine chemistry scheme.

    "*...suggesting that there are still uncertainties in other aspects of our understanding of the iodine system such as $HIO_3$ (He et al., 2021; Huang et al., 2022) and the photolysis scheme currently used for higher iodine oxides (Sherwen et al., 2016). Work to further understand the atmospheric chemistry of iodine is still required if we are to have confidence in the predictions of our models.*"

Indeed, looking at the ozone comparison, the match seems to be worse for most stations. The model was run from 2019 to 2021, but the observations are from 2014. Why was this period used for the model run? What are the possible reasons for no improvement? How much is inter-annual meteorological variability expected to affect the comparison? Why not use a climatology for the GAW stations?

- The iodide field used to drive the iodine emissions (Sherwen et al 2019) is a climatology and therefore is not subject to interseasonal variability. There are still large uncertainties in the emissions of inorganic iodine with this work only focusing on inorganic chemistry and not the role of surfactants. Additionally there remain uncertainties in the atmospheric chemistry mechanism of iodine. The authors have included a discussion of uncertainties on lines (401-404) and clarified that the iodide used by the model is a climatology on line 365.

- We do not expect a significant impact from inter-annual meteorological variability particularly when comparing monthly mean values.

Please include IO observations from the Indian Ocean (Mahajan et al., 2019a, b; Inamdar et al., 2020), the Pacific (Takashima et al., 2022) and the Malaspina circumnavigation cruise (Prados-Roman et al., 2015) for model comparison and validation. Observations from the Arctic Ocean are also missing in this comparison (Benavent et al., 2022). The authors also do not include coastal observations, but this is most likely to avoid macroalgae emissions – please state this clearly. In addition, ground-based observations in the free troposphere should also be compared with those of (Dix et al., 2013) (Volkamer et al., 2015), considering that the model has organic emissions.

- Additional observations of marine tropospheric IO have been included in the comparison between the model and observations in Figure 14 (formerly Figure 12)
- Lines 396-398 has been included to clearly state that coastal observations are excluded.

> "*This comparison only considers open ocean observations, coastal observations of IO have large influences from macro-algae emissions (Saiz-Lopez and Plane 2004) which are not included in this model.*"

- This work has not altered the existing organic emissions and is primarily focused on the marine boundary layer where inorganic iodine emissions have the greatest impact on tropospheric chemistry due to their short lifetime. Wang et al. 2021 (10.5194/acp-21-13973-2021) have previously made comparisons to free troposphere observations.

Initial work on the effect of iodine on ozone has been ignored. For example, the first large-scale modelling efforts by (Saiz-Lopez et al., 2012) have not been cited, and nor has the work that identified the atmospheric chemistry of iodine through its impact on ozone loss and HOx changes (Vogt et al., 1999; Alicke et al., 1999; Allan et al., 2000; Saiz-Lopez and Plane, 2004; Bloss et al., 2005). Even the recent publications showing the large-scale impacts of short-lived halogens on ozone, aerosols, HOx and aerosols have not been cited (Saiz-Lopez et al., 2023).

- The authors thank the reviewer for the additional references and these have been included with lines 28-30 updated to read as follows

> "*Photochemical cycling of iodine in the atmosphere leads to efficient chemical loss of $O_3$, perturbs $HO_x$ (Vogt et al., 1999; Alicke et al., 1999; Allan et al., 2000; Bloss et al., 2005) and along with other*"

> > *short-lived halogens emitted from the ocean surface have a substantial indirect impact on climate (Saiz-Lopez et al., 2023)*"

> - Line 31-32 updated to the following

> > "*The dominant loss route is IO + HO$_2$ to return to HOI, which on photolysis leads to net loss of O$_3$ (Sommariva et al., 2012; Saiz-Lopez et al., 2012)*"

> - Line 396-398 updated to

> > "*This comparison only considers open ocean observations, coastal observations of IO have large influences from macro-algae emissions (Saiz-Lopez and Plane 2004) which are not included in this model.*"

Line 38: 'particle' instead of 'partial'

> - The text of line 42 has been updated to now correctly read "*Recent work has also supports atmospheric iodine playing an important role in particle formation in the MBL…*"

Line 43: The contribution of inorganic iodine emissions to stratospheric iodine loading is very small, considering its short lifetime. Please make it clear that this is due to organic iodine emissions.

> - The authors thank the reviewer for this comment and have clarified the statement on line 47/48 to now read

> > "*Recent observations show that approximately 0.7 ppt of reactive iodine species are injected into the stratosphere, largely in the form of longer lived organic iodine species and particulate iodine (Koenig et al. 2020).*"

Line 61: There is field proof of this – parameterised inorganic iodine emissions had to be reduced to 40% to match observations in the Indian Ocean (Mahajan et al., 2021).

> - The authors thank the reviewer for this additional reference and have included this in the introduction on line 74-75

> > "*Modelling studies of IO in the Indian Ocean needed to reduce inorganic iodine emissions by 40% to reasonably match cruise-based observations from the region*"

Line 66: and decrease in sea-ice extent leading to more exposed seawater in the Arctic.

> - The authors acknowledge there is a complex relationship between climate and iodine emissions, however, this is beyond the scope of this work, however could warrant future research.

Line 195: The authors mention that the other processes for the depletion of I- are likely to be minor – however, considering significant emissions of iodocarbons from the ocean surface, would I- contribute towards this, and if so, how can this be considered?

- Laboratory studies using SML samples found that emission of organic iodine was a negligible component of total iodine emissions, this has been included in the introduction on line 70/71

"*Laboratory studies of ozonised SML samples found that volatile organic iodine emissions were a negligible fraction of total iodine emissions (Tinel et al., 2020)*"

Line 224: 'by ~170% when using the rate coefficient from Brown et al. (2023)' – not sure how this calculation has been made. From the figure, the change is approximately 0.7/0.3 (an increase of a factor of 2.3).

- This figure and discussion of it (lines 253 & 255) have been updated to correctly reflect the data on the plot and the updates made to the published rate constant since the review process began.

Figure 6: The x-axis is wind speed, but the caption mentions SST.

- Corrected figure caption to now correctly reference the x-axis being wind speed rather than SST.

Figure 11: Caption – state what this is change relative to.

- Figure 11 caption updated to state that this is a change between the "base" case model using the old inorganic iodine emissions from the ocean and the new emissions presented in this work

*"Annual mean mixing ratio of IO (top) using new inorganic iodine emissions, absolute change (middle) and percentage change (bottom) in the annual mean atmospheric IO from implementing the new inorganic iodine emissions relative to the "base" model case which used oceanic iodine emissions from Carpenter et al. 2013."*

Figure 7: The x-axis is wind speed, but the caption mentions SST. Add space between the $O_3$ concentration number and the unit.

- Corrected figure caption to now correctly reference the x-axis being wind speed rather than SST.

Figure 13: Please make clear whether this is the difference in ozone due the parameterisations or due to iodine chemistry.

- Figure 13 caption updated to specifically state that this is the change in ozone due to the oceanic iodine emissions used

*"Absolute and percentage change in surface $O_3$ (a and c), absolute and percentage change in zonal $O_3$ (b and d) due to changing the Carpenter et al. 2013 inorganic iodine emissions from the ocean to the equations presented in this work. …"*

---

## Author Response (AR2)

The authors thank the editor for their comments and have provided additional clarification and discussion on La Niña and model comparisons below. All page and line numbers referenced in this response correspond to the tracked changes version of the manuscript.

Dear Ryan J. Pound

Thank you for the revised submission and carefully handling the referee's comments. A referee insisted on clarifying the impact of the simulations covering a LaNina period, pointing out that sea surface temperatures might differ from those of other years. While it is obvious and clearly stated throughout the manuscript that the GEOS-Chem classic model results cover the 2 years 2020/21, could you clarify the impact of this choice of years for me in one last iteration? I decided not to send the manuscript to the referee but to handle this last step as editor to speed up the process.

It is difficult for me to judge how exactly LaNina years differ from other years. The discussion around Figures 12 and 13 showed "higher emissions at higher latitudes and a decrease in emissions from warmer, tropical waters." The manuscript identifies temperature as a key parameter determining iodine emissions (Sections, 3,4,7); am I correct? Which temperatures did the model derive for the high and low latitutes and how much do they differ from "normal" years. Could you comment on whether you'd expect similar large changes in other years, or in other words, are the findings particularly enhanced due to anomaly temperatures in these years?

You are correct. Temperature is a key parameter in determining the inorganic iodine emissions and a key development in this work compared to Carpenter et.al 2013. This comparison shows the temperatures from the MERRA2 meteorology used to drive the GEOS-Chem model and the difference between the 2020+2021 average to the previous decade (at the 4°x5° resolution used for the global modelling results presented in this work).

[Figure]

Figure 10(d) in the manuscript shows the temperature dependence of the inorganic iodine emissions. Comparing the temperature anomaly for the 2020/2021 period to this figure, the difference in sea surface temperatures in these La Niña years does not have a significant impact on our conclusions and we would expect to see similar results to those presented in this manuscript in other years.

The authors have included additional discussion of this in the manuscript on lines 359-362:

*"Although the 2020-2021 period is within a La Niña event, a change in temperature of 1K changes total inorganic iodine emissions by ~ 3 % (figure 10). As such temperature variations due to ENSO are likely to result in changes in the inorganic iodine concentrations of less than 10% locally, and likely less globally. "*

Why were modeling results compared with observations at different times? I assume it is impossible to run the model over longer times for time reasons. My concern is that temperatures (and other parameters) might differ between the years, so one would not necessarily expect a good agreement. In this respect, please explain the "equivalent model" in line 396 in more detail – I first misunderstood this as temperature and other parameters (wind speed) being identical, similar to the comparison with the lab experiments. A short sentence would help a lot, thank you. Could you also comment, if not done above, how much the modeled meteorology is affected by the strong La Nina and whether this can have impact on the simulated iodine emissions.

As discussed above, even the relatively large SST change in the 2020+2021 La Nina period, compared with the previous decade, results in only a small difference in the model predictions for inorganic iodine emissions.  As such we do not expect differing years to be a large source of uncertainty in these model-observation comparisons.

The text on lines 382-383 has been updated to read
*"...compares published observations of average daytime surface IO mixing ratios to the model predictions from the corresponding day of the year during the simulated period."*

I also understood that both model results are based on identical temperature and wind speed, that is, meteorology – the difference of interest here is the chemistry. More for curiosity, can we learn more from Figure 16? There appear to be time periods with better and worse agreement in each location.

You are correct, the meteorology is identical in the "old" and "new" model results, along with all emissions (excluding inorganic oceanic iodine emissions) and the atmospheric chemistry scheme. These two scenarios show the sensitivity of modelled surface $O_3$ to the change in oceanic iodine emissions.  While there are times of better or worse agreement between the model and the measurements, it is unlikely that model failure is determined solely by the iodine emissions. Overall, uncertainties in the chemistry, transport and deposition, together with errors and uncertainties in the emission of other species (NOx, VOCs, halogens, particulate) will combine to provide the overall error profile. This additional discussion has been included on lines 430-433

*"While there are times of better or worse agreement between the model and observations at all locations presented in figure 16, model failure is likely not strongly influenced by year-to-year variability in the iodine emissions. Overall, uncertainties in the chemistry, transport and deposition of iodine species, together with errors and*

*__uncertainties in the emission of other species (NO$_x$, VOC's, halogens, particulate) will combine to provide the overall error profile."__*

In summary, could you elaborate on the possible impact of LaNina on the results and, more generally, on the use of different years for the model observation comparison? Or, make the statement clearer that the results do not reflect possible impacts of LaNina.

Thank you very much, I hope the comments are clear, kind regards, Thorsten Bartels-Rausch